# AMDP: An Adaptive Detection Procedure for False Discovery Rate Control in High-Dimensional Mediation Analysis

**Jiarong Ding**
School of Mathematics and Statistics
Xi'an Jiaotong University
djr9901@stu.xjtu.edu.cn

**Xuehu Zhu**[*]
School of Mathematics and Statistics
Xi'an Jiaotong University
zhuxuehu@xjtu.edu.cn

## Abstract

High-dimensional mediation analysis is often associated with a multiple testing problem for detecting significant mediators. Assessing the uncertainty of this detecting process via false discovery rate (FDR) has garnered great interest. To control the FDR in multiple testing, two essential steps are involved: ranking and selection. Existing approaches either construct p-values without calibration or disregard the joint information across tests, leading to conservation in FDR control or non-optimal ranking rules for multiple hypotheses. In this paper, we develop an adaptive mediation detection procedure (referred to as "AMDP") to identify relevant mediators while asymptotically controlling the FDR in high-dimensional mediation analysis. AMDP produces the optimal rule for ranking hypotheses and proposes a data-driven strategy to determine the threshold for mediator selection. This novel method captures information from the proportions of composite null hypotheses and the distribution of p-values, which turns the high dimensionality into an advantage instead of a limitation. The numerical studies on synthetic and real data sets illustrate the performances of AMDP compared with existing approaches.

## 1 Introduction

Mediation analysis is regarded as a prevalent tool to dissect a mediation relationship between exposures and outcomes, and it has been widely applied in different fields, such as epidemiology [37], public health [21], policy evaluation [1], social sciences [25], neuroscience [10], and many others. Baron et al. [4] provided the basis for the advance of mediation analysis. They proposed a conventional regression-based approach, commonly referred to as the causal steps method, to examine the logical relationships among exposure, mediator, and outcome variables linking in a causal chain. While the causal steps method established necessary conditions for causal inference, it did not provide a joint test of the indirect effect of exposure on the outcome through a mediator, as recommended by MacKinnon et al. [32]. Thus, MacKinnon et al. [32] investigated the normality-based Sobel's test [38] as a means to detect mediation effects under the framework of the product-coefficient method, the product of the exposure-mediator and mediator-outcome effects. However, as highlighted by MacKinnon et al. [31], the distribution of product coefficients is not normally distributed, leading the Sobel's test being overly conservative. Hence, MacKinnon et al. [32] proposed the joint significance test (also name as the MaxP test) to alleviate this conservatism. Nonetheless, the joint significance test still suffers from low statistical power, as it overlooks the impact of the composite null structure in mediation analysis [33, 42].

---

[*]Corresponding author

37th Conference on Neural Information Processing Systems (NeurIPS 2023).

To address the aforementioned issues, Taylor et al. [42] recommended the utilization of the distribution of the product method or bootstrapping as alternative procedures. These methods had been shown to exhibit higher power while still maintaining reasonable control over the Type I error rate. In the study of Kamukama et al. [27], a z-value based Sobel's test was introduced to investigate the mediation effect of competitive advantage on the relationship between intellectual capital and financial performance. Nuijten et al. [36] proposed an approach based on Bayesian models for testing the presence of an indirect effect. More recently, Zhang [46] developed two data-adaptive tests that outperform both Sobel's test and joint significance tests.

Despite these advances, a further challenge is that the methods mentioned above typically assume a low-dimensional mediator, whereas the development of high-throughput technologies promotes a growing need for research dealing with high-dimensional data. The multiple testing problem arising from high-dimension mediation analysis is aimed at identifying the relevant mediators that explain the effect of exposure on an outcome. Differ from a single test, multiple testing approaches are aimed to handle simultaneous testing, and often use the false discovery rate (FDR) to measure the uncertainty of detecting process [7], that is,

$$\text{FDP} = \frac{\#\left\{j : j \in \Delta_0, j \in \widehat{\Delta}\right\}}{\#\{j \in \widehat{\Delta}\} \vee 1}, \quad \text{FDR} = \mathbb{E}[\text{FDP}], \quad (1)$$

where $j$ represents the index of mediator, $\Delta_0$ is the index set of the null mediators, $\widehat{\Delta}$ is the index set of selected mediators.

The control of FDR in multiple hypothesis testing consists of two primary steps: ranking and selection. In the initial step, a ranking statistic is calculated to evaluate the significance of each test, resulting in the ranking of hypotheses. The subsequent step is to maximize the selection set based on the established ranking order from the first step, while simultaneously maintaining the FDR at the target level. Although the p-value generated from the single hypothesis testing method can be considered as a ranking statistic and further combined with the BH procedure [7] for FDR control, there are still some limitations inherent to this approach. Firstly, the single p-value without calibration may be conservative and lead to excessive conservation in FDR control. Secondly, in the absence of joint information across multiple hypotheses, the ranking statistics based on the p-value may not be optimal.

To overcome these limitations, recent advancements in mediation analysis attempt to construct calibrated test statistics that account for the composite nature of the null hypothesis, and further achieve FDR control in multiple testing. For example, Dai et al. [12] developed a JS-mixture procedure, which utilizes the maxP statistics [32] as the ranking statistics and corrects the conservatism in the joint significance test by estimating the mixture distribution of p-values. Although JS-mixture sharply achieves FDR control, it is still underpowered for the reason that the maxP statistics does not account for the distribution information of two-dimension p-values under different hypotheses. Furthermore, Liu et al. [30] proposed a procedure called DACT focusing on constructing calibrated p-values for each single test by combining information across large-scale tests. Specially, DACT estimates the proportions of sub-null hypotheses and generates weighted p-values accordingly. However, DACT may suffer from underpowered performance in certain situations due to the following reasons. Intuitively, it ignores the alternative hypothesis information in its weighting scheme. On a deeper level, it does not fully consider the distributed information of two-dimensional p-values, limiting its ability to leverage valuable insights for improved power.

In this paper, we propose an adaptive mediation detection procedure (AMDP) for identifying relevant signals in high-dimensional mediation analysis. Our main contributions are summarized below:

- AMDP utilizes a two-dimensional p-value based local FDR as a test statistic, allowing for the comprehensive utilization of structural information of large-scale tests. Additionally, it determines the optimal rule for the order of selecting mediators.
- We establish theoretical results showing that AMDP enables asymptotic control of the FDR for selected mediators using the estimated local FDR.
- We reveal the critical importance of information retention in the ranking step for achieving optimal statistical power by discussing the limitations of the ranking statistics used in existing methods and conducting comparisons with AMDP.
- We empirically demonstrate the effectiveness of AMDP on synthetic and real data sets. Simulation results confirm the validity of our approach, and an application to a prostate

cancer dataset illustrates its satisfactory performance in identifying CpG methylation sites that mediate between risk SNPs and gene expression.

The remainder of this paper is organized as follows: Section 2 formally introduces an optimal ranking rule based on AMDP along with an estimator of local FDR. We theoretically prove the ability of AMDP in controlling the FDR while mimicking the optimal power. Section 3 presents the simulation studies to evaluate the performance of AMDP. In Section 4, we demonstrate the practical utility of AMDP by applying it to the prostate cancer dataset in TCGA 2015. We conclude the paper in Section 5. The technical proofs and additional discussion are postponed to Appendix.

## 2 AMDP: an optimal multiple testing procedure for FDR control

Let $X$ be the exposure (independent variable), $\{M_1, \cdots, M_J\}$ be candidate mediators, and $Y$ be the outcome (dependent variable). In the context of mediation analysis in the genome-wide association studies (GWASs), $X$ often refers to the single nucleotide polymorphisms (SNPs), $M$ corresponds to DNA methylation, and $Y$ pertains to gene expression or a risk of disease. As stated by Baron et al. [4], the mediation relationship can be expressed by the following models:

$$
\begin{aligned}
\mathbb{E}\left(M_j \mid X\right) &= \alpha_{0j} + \alpha_j X, \\
\mathbb{E}\left(Y \mid M_j, X\right) &= \beta_{0j} + \beta_j M_j + \beta_{1j} X,
\end{aligned}
\tag{2}
$$

where $\alpha_j$ denotes the effect of $X$ on $M_j$, $\beta_j$ represents the effect of $M_j$ on $Y$, totally the product of $\alpha_j$ and $\beta_j$ represents indirect effect of $X$ on $Y$. $\beta_{1j}$ is the direct effect of $X$ on $Y$ with the $M_j$ being fixed. We assume that there are no unmeasured confounding variables, also known as the sequential ignorability assumption [25, 43]. Any confounders can be adjusted by additional covariates [12], and such an adjustment is omitted in model (2) for simplification.

Testing whether $\{M_1, \ldots, M_J\}$ plays an intermediary role in the causal path from $X$ to $Y$ in (2) can be transformed into a multiple testing problem:

$$
H_{0j} : \alpha_j \beta_j = 0 \text{ versus } H_{1j} : \alpha_j \beta_j \neq 0.
\tag{3}
$$

The above composite hypothesis can be decomposed into four disjoint cases as follows:

$$
\begin{aligned}
\text{Case 1, } \mathrm{H}_{00,j} : & \quad \alpha_j = 0 \text{ and } & \beta_j = 0, \\
\text{Case 2, } \mathrm{H}_{01,j} : & \quad \alpha_j = 0 \text{ and } & \beta_j \neq 0, \\
\text{Case 3, } \mathrm{H}_{10,j} : & \quad \alpha_j \neq 0 \text{ and } & \beta_j = 0, \\
\text{Case 4, } \mathrm{H}_{11,j} : & \quad \alpha_j \neq 0 \text{ and } & \beta_j \neq 0,
\end{aligned}
$$

where Case 1-3 represents the composite null hypothesis, and Case 4 is the alternative hypothesis. A rejection of $H_{0j}$ indicates the presence of a mediation effect by $M_j$. In this paper, the $p$-values for testing $\alpha_j = 0$ and $\beta_j = 0$ are respectively denoted as $p_{1j} = 2\{1 - \Phi(\mid \hat{\alpha}_j \mid / \hat{\sigma}_{\alpha_j})\}$ and $p_{2j} = 2\{1 - \Phi(\mid \hat{\beta}_j \mid / \hat{\sigma}_{\beta_j})\}$, where $\hat{\alpha}_j$, $\hat{\beta}_j$, $\hat{\sigma}_{\alpha_j}$ and $\hat{\sigma}_{\beta_j}$ are the least squares estimators based on the models (2). Under the sequential ignorability assumption for mediation analysis, $p_{1j}$ and $p_{2j}$ are independent [30].

### 2.1 Optimal rejection region under the four-group model

To provide an optimal ranking guideline, we consider a pair of p-values $p_j = (p_{1j}, p_{2j})$ under the empirical null inference framework [17]. Let $H_{00,j} = 1$ if Case 1 holds, $H_{01,j} = 1$ if Case 2 holds, $H_{10,j} = 1$ if Case 3 holds, and $H_{11,j} = 1$ otherwise. Assume the conditional density of $p_j$ follows

$$
H_j = \begin{cases}
H_{00,j} \sim \text{Bernoulli}\{\pi_{00}\}, \\
H_{01,j} \sim \text{Bernoulli}\{\pi_{01}\}, \\
H_{10,j} \sim \text{Bernoulli}\{\pi_{10}\}, \\
H_{11,j} \sim \text{Bernoulli}\{\pi_{11}\},
\end{cases}
\quad p_j \mid H_j \sim \begin{cases}
f_{00}(p) & \text{if } H_{00,j} = 1, \\
f_{01}(p) & \text{if } H_{01,j} = 1, \\
f_{10}(p) & \text{if } H_{10,j} = 1, \\
f_{11}(p) & \text{if } H_{11,j} = 1.
\end{cases}
\tag{4}
$$

with $p = \left(p^{(1)}, p^{(2)}\right) \in \mathbb{R}^2$. $\pi_{00}, \pi_{01}, \pi_{10}$, and $\pi_{11}$ represents the proportions of $H_{00,j}, H_{01,j}, H_{10,j}$, and $H_{11,j}$, respectively. It follows that $\pi_{00} + \pi_{01} + \pi_{10} + \pi_{11} = 1$ due to the disjoint nature of the composite hypothesis.

Then, the density function of $p_j$ follows the following four-group model, which can be considered as a variant version of the random mixture model [17].

$$p_j \sim f(p) = \pi_{00}f_{00}(p) + \pi_{01}f_{01}(p) + \pi_{10}f_{10}(p) + \pi_{11}f_{11}(p). \tag{5}$$

Under the four-group model (5), the local FDR [18, 19] is defined as

$$\mathrm{fdr}(p) = \mathbb{P}\left(H_{00,j} \cup H_{01,j} \cup H_{10,j} = 1 \mid p_j = p\right) = \frac{\pi_{00}f_{00}(p) + \pi_{01}f_{01}(p) + \pi_{10}f_{10}(p)}{f(p)}. \tag{6}$$

It refers to the posterior probability that a hypothesis is null, given its corresponding p-value.

Before delving into the optimal ranking guidelines, we introduce several key definitions relevant to this objective. For any rejection region $S \in [0,1]^2$, we define the global FDR as

$$\mathrm{gFDR}(S) = \mathbb{P}(H_{00} \cup H_{01} \cup H_{10} = 1 \mid p_j \in S), \tag{7}$$

where $H_{00}, H_{01}$, and $H_{10}$ are composite null hypothesis. The power is defined as

$$\mathrm{Power}(S) = \mathbb{P}\{p_j \in S \mid H_{11} = 1\}. \tag{8}$$

In the ranking step, the primary objective is to establish an optimal ranking rule that accurately reflects the significance order of the tests, while adhering to the optimality goal set in the selection step. Under the Neyman-Pearson framework [35], this optimality goal entails maximizing power while simultaneously controlling the global FDR at a targeted level of $\alpha$. This process can be formulated as a constrained optimization problem, i.e.

$$\max_S \mathrm{Power}(S) \quad \text{subject to } \mathrm{gFDR}(S) \leqslant \alpha. \tag{9}$$

The optimal rule under the two-group model has been extensively studied in the literature. Researchers have proposed various methods for optimal decision-making based on different frameworks [5, 9]. Our optimality goal shares similarities with the work of Lei et al. [28]. They had demonstrated that, under Bayes rule, the optimal rejection thresholds are the level surfaces of local FDR. We extend this insight to p-values in two dimensions, and define the form of the rejection region as $S(\zeta) = \{p : \mathrm{fdr}(p) \leq \zeta\}$. A detailed and comprehensive explanation of this concept is provided in Theorem 1.

**Theorem 1.** *Assume that*

  (i) *$f_{00}(p)$, $f_{01}(p)$, $f_{10}(p)$, and $f_{11}(p)$ are continuous;*

  (ii) *$\nu(p : \mathrm{fdr}(p) = t) = 0$ for any $t \in (0,1]$, where $\nu$ is a Lebesgue-Stieltjes measure on the two-dimensional Borel space $(\mathcal{R}^2, \mathcal{B}^2)$.*

*Then, for any given global FDR level $\alpha$, there exists a unique value $\zeta^\star$ such that $S(\zeta^\star)$ is the solution of the constrained optimization problem in (9). And the local FDR involved in $S(\zeta^\star)$ corresponds to the optimal ranking rule.*

**Remark 1.** *Genovese et al. [20] have shown that under weak conditions, $\mathrm{gFDR} = \mathrm{FDR} + O(\frac{1}{\sqrt{J}})$, where J represents the number of mediators. Hence, controlling $\mathrm{gFDR}$ and $\mathrm{FDR}$ are asymptotically equivalent as the number of mediators J tends to infinity. A similar result supporting this equivalence was also obtained by Storey [40].*

## 2.2 The estimator of local FDR

From Theorem 1, we have established that the optimal ranking rule under the Neyman-Pearson framework is the local FDR. However, it is worth noting that the discussions in Section 2.1 are based on the assumption that the distribution of p-values and proportions of the composite hypothesis are known. In the following, we emphasize that our results still hold if the local FDR can be consistently estimated. Assuming that $f_{00} \equiv 1$ (p-values follow uniform distribution in Case 1). The estimation of $\mathrm{fdr}(p)$ can be divided into three parts: (i) The proportions of the composite null hypothesis $\pi_{00}, \pi_{01}, \pi_{10}$; (ii) The mixture density $f(p)$; (iii) The densities of the composite null hypothesis $f_{01}(p), f_{10}(p)$.

Motivated from Storey et al. [41], $\pi_{01}, \pi_{10}$, and $\pi_{00}$ can be estimated as follows:

$$\hat{\pi}_{0\cdot}(\lambda) = \frac{\sum I\left(p_{1j} > \lambda\right)}{J(1-\lambda)}, \quad \hat{\pi}_{\cdot 0}(\lambda) = \frac{\sum I\left(p_{2j} > \lambda\right)}{J(1-\lambda)}, \quad \hat{\pi}_{00}(\lambda) = \frac{\sum I\left(p_{1j} > \lambda, p_{2j} > \lambda\right)}{J(1-\lambda)^2}, \tag{10}$$

where $\hat{\pi}_{0\cdot}(\lambda)$ denotes the estimator of the proportion of null $p_{1j}$, $\hat{\pi}_{\cdot0}(\lambda)$ denotes the estimator of the proportion of null $p_{2j}$. $I(\cdot)$ is an indicator function, and $\lambda \in [0,1)$ is a tuning parameter. In practice, there is a bias versus variance tradeoff for choosing a suitable $\lambda$. Further research on selecting an appropriate value of $\lambda$ is detailed in the Appendix A. Following that

$$\hat{\pi}_{01}(\lambda) = \hat{\pi}_{0\cdot}(\lambda) - \hat{\pi}_{00}(\lambda), \quad \hat{\pi}_{10}(\lambda) = \hat{\pi}_{\cdot0}(\lambda) - \hat{\pi}_{00}(\lambda). \tag{11}$$

Next, we turn to the estimation of $f(p)$. For this purpose, we employed an adaptation of the beta kernel function proposed by Chen [11]. This choice is made considering the fact that $p = (p^{(1)}, p^{(2)})$ falls within $[0,1]^2$. The beta kernel function allows for a flexible and smooth estimation of $f(p)$, providing a suitable estimation approach for our analysis. Our beta kernel estimator is:

$$\hat{f}(p) = \hat{f}(p^{(1)}, p^{(2)}) = J^{-2} \left( \sum_{j=1}^{J} K^{\star}_{p^{(1)},b}(p_{1j}) \right) \left( \sum_{j=1}^{J} K^{\star}_{p^{(2)},b}(p_{2j}) \right), \tag{12}$$

where $K^{\star}_{p,b}$ is a boundary beta kernel defined as

$$K^{\star}_{p,b}(t) = \begin{cases} K_{p/b,(1-p)/b}(t) & \text{if } p \in (2b, 1-2b), \\ K_{\rho(p,b),(1-p)/b}(t) & \text{if } p \in [0, 2b], \\ K_{p/b,\rho(1-p)}(t) & \text{if } p \in [1-2b, 1], \end{cases}$$

$K_{u,v}$ be the density function of a $\text{Beta}(u,v)$ random variable, $b$ is a smoothing parameter, and $\rho(p,b) = 2p^2 + 2.5 - \sqrt{4p^4 + 6p^2 + 2.25 - p^2 - p/b}$.

In the context of mediation analysis, the density of p-values follows a mixture distribution, as indicated in (5). This mixture distribution involves three distinct types of null hypotheses: $H_{01}$, $H_{10}$, and $H_{00}$. Distinguishing between $H_{01}$ and $H_{10}$, as well as obtaining accurate estimators for $f_{01}(p)$ (corresponding to $H_{01}$) and $f_{10}(p)$ (corresponding to $H_{10}$) is indeed a challenging task. Motivated by the knockoff method [2], we consider leveraging the symmetry property of p-values under the composite null hypothesis to tackle this issue. Before diving into the details of utilization of the symmetry property for estimating $f_{01}(p)$ and $f_{10}(p)$, we introduce some essential notations and assumptions.

Denote $\Delta_{00}, \Delta_{01}$, and $\Delta_{10}$ as the index set of the null mediators under the composite null hypothesis $H_{00}, H_{01}$, and $H_{10}$, respectively. We define the region $D$ as $[0, 0.5)^2$, with its symmetric regions as follows: $\tilde{D}_{01} = [0.5, 1] \times [0, 0.5)$, $\tilde{D}_{10} = [0, 0.5) \times [0.5, 1]$, and $\tilde{D}_{00} = [0.5, 1]^2$. The assumptions are given as follows.

**Assumption 1.** *For $j \in \Delta_{00}$, the sampling distribution of $p_j$ is symmetric about $p^{(1)} = 0.5$ and $p^{(2)} = 0.5$; For $j \in \Delta_{01}$, the sampling distribution of $p_j$ is symmetric about $p^{(1)} = 0.5$; For $j \in \Delta_{10}$, the sampling distribution of $p_j$ is symmetric about $p^{(2)} = 0.5$.*

**Assumption 2.** *The symmetric regions of $D$ satisfy: (i) For $p \in \tilde{D}_{00}$, $\lim_{n \to \infty} f_{11}(p) = 0$, $\lim_{n \to \infty} f_{01}(p) = 0$, $\lim_{n \to \infty} f_{10}(p) = 0$; (ii) For $p \in \tilde{D}_{01}$, $\lim_{n \to \infty} f_{11}(p) = 0$ and $\lim_{n \to \infty} f_{10}(p) = 0$; (iii) For $p \in \tilde{D}_{10}$, $\lim_{n \to \infty} f_{11}(p) = 0$ and $\lim_{n \to \infty} f_{01}(p) = 0$.*

Assumption 1 is only required for the null mediators. It indicates that at least one of $p_{1j}$ and $p_{2j}$ follows a uniform distribution under the composite null hypothesis. Assumption 2 holds for any reasonable p-value. Since a non-null p-value should fall within $[0, 0.5)$, we can infer that as the sample size $n$ tends to infinity, the probability of p-values under alternatives falling within $[0.5, 1]$ approaches zero. Additional explanations on Assumptions 1-2 are detailed in the Appendix D.2.

Remarkably, under Assumption 1, we can decompose $f_{10}(p)$ and $f_{01}(p)$ as follows:

$$f_{10}(p) = f_{1\cdot}(p^{(1)}) \cdot f_{\cdot0}(p^{(2)}) = f_{1\cdot}(p^{(1)}), \quad f_{01}(p) = f_{0\cdot}(p^{(1)}) \cdot f_{\cdot1}(p^{(2)}) = f_{\cdot1}(p^{(2)}), \tag{13}$$

where $f_{0\cdot}(p^{(1)}) = f_{\cdot0}(p^{(2)}) = 1$. This decomposition allows us to transform the problem into estimating $f_{1\cdot}(p^{(1)})$ and $f_{\cdot1}(p^{(2)})$, representing the marginal probability density of $p_{1j}$ and $p_{2j}$ under alternatives, respectively. Assumption 2 provides the inspiration to utilize the symmetric regions about $D$ to address this estimation task, that is,

$$k_1(p^{(1)}) = \frac{\pi_{00} + \pi_{10}f_{1\cdot}(p^{(1)})}{\pi_{00} + \pi_{10}}, \quad k_2(p^{(2)}) = \frac{\pi_{00} + \pi_{01}f_{\cdot1}(p^{(2)})}{\pi_{00} + \pi_{01}}, \tag{14}$$

where $k_1(p^{(1)})$ denotes the density of $p_{1j}$ under $H_{10}$ and $H_{00}$, and $k_2(p^{(2)})$ denotes the density of $p_{2j}$ under $H_{01}$ and $H_{00}$.

We apply the beta kernel function to $p_{1j}$ in region $\tilde{D}_{10}$ and $\tilde{D}_{00}$, as well as to $p_{2j}$ in region $\tilde{D}_{01}$ and $\tilde{D}_{00}$, to get the estimation of $k_1(p^{(1)})$ and $k_2(p^{(2)})$ as

$$\hat{k}_1(p^{(1)}) = J_1^{-1} \sum_{j=1}^{J_1} K^{\star}_{p^{(1)},b}(p_{1j}), \quad \hat{k}_2(p^{(2)}) = J_2^{-1} \sum_{j=1}^{J_2} K^{\star}_{p^{(2)},b}(p_{2j}), \tag{15}$$

where $J_1$ is the number of p-values in region $\tilde{D}_{10}$ and $\tilde{D}_{00}$, $J_2$ is the number of p-values in region $\tilde{D}_{01}$ and $\tilde{D}_{00}$. Combining the estimators in (10)-(11), (13)-(15), we obtain the estimation of $f_{1\cdot}(p^{(1)})$ and $f_{\cdot1}(p^{(2)})$ as:

$$\hat{f}_{1\cdot}(p^{(1)}) = \frac{(\hat{\pi}_{00} + \hat{\pi}_{10})\hat{k}_1(p^{(1)}) - \hat{\pi}_{00}}{\hat{\pi}_{10}}, \quad \hat{f}_{\cdot1}(p^{(2)}) = \frac{(\hat{\pi}_{00} + \hat{\pi}_{01})\hat{k}_2(p^{(2)}) - \hat{\pi}_{00}}{\hat{\pi}_{01}}. \tag{16}$$

Therefore, the local FDR estimator is derived as:

$$\widehat{\mathrm{fdr}}(p) = \frac{\hat{\pi}_{00}f_{00}(p) + \hat{\pi}_{01}\hat{f}_{01}(p) + \hat{\pi}_{10}\hat{f}_{10}(p)}{\hat{f}(p)}. \tag{17}$$

## 2.3 Asymptotic FDR control

In this section, we present a selection strategy for the second step. The primary goal of our proposed selection strategy is to maximize power while simultaneously controlling the FDR based on the established ranking order from the first step, i.e., finding the optimal threshold $\zeta^{\star}$ of the optimization problem (9). However, determining such an optimal threshold $\zeta^{\star}$ is a challenging task, as it involves decision-making based on the estimation of global FDR. To address this challenge effectively, we propose a data-driven strategy. Based on the notations in Section 2.2, the form of the FDP and FDR are given by

$$\mathrm{FDP}(\zeta) = \frac{\#\left\{j : j \in \Delta_{00} \cup \Delta_{01} \cup \Delta_{10}, j \in \widehat{\Delta}\right\}}{\#\{j : j \in \widehat{\Delta}\} \vee 1} \text{ and } \mathrm{FDR}(\zeta) = \mathrm{E}(\mathrm{FDP}(\zeta)), \tag{18}$$

where $\widehat{\Delta}$ is the index set of selection in rejection region $\widehat{S}(\zeta) = \{p : \widehat{\mathrm{fdr}}(p) \leq \zeta\}$, i.e., $j \in \widehat{\Delta}$ when $p_j \in \widehat{S}(\zeta)$, the denominator of $\mathrm{FDP}(\zeta)$ represents the total number of rejections and the numerator represents the number of false positives.

In mediation analysis, accurately estimating the number of false discoveries in (18) poses a challenge, since the rejection region $\widehat{S}(\zeta)$ comprises a mixture of four distinct types of hypotheses. However, we can draw inspiration from Assumptions 1 and 2 to leverage the symmetry property of p-values under the composite null hypothesis to estimate the number of false positives. We define the symmetric regions of $\widehat{S}$ as $\tilde{S}_{01} = \{(1 - p^{(1)}, p^{(2)}) : \widehat{\mathrm{fdr}}(p) \leq \zeta\}$, $\tilde{S}_{10} = \{(p^{(1)}, 1 - p^{(2)}) : \widehat{\mathrm{fdr}}(p) \leq \zeta\}$, $\tilde{S}_{00} = \{(1 - p^{(1)}, 1 - p^{(2)}) : \widehat{\mathrm{fdr}}(p) \leq \zeta\}$. It's noteworthy that the rejection region $\widehat{S}$ is a subset of the region $D$ defined in Section 2.2, and its symmetric regions, $\tilde{S}_{01} \subseteq \tilde{D}_{01}$, $\tilde{S}_{10} \subseteq \tilde{D}_{10}$, and $\tilde{S}_{00} \subseteq \tilde{D}_{00}$. Indeed, Assumptions 1 and 2 provide us with an approximation of the number of false positives in (18):

$$\#\left\{j \in \Delta_{01} \cup \Delta_{00} : p_j \in \widehat{S}(\zeta)\right\} \approx \#\left\{j \in \Delta_{01} \cup \Delta_{00} : p_j \in \tilde{S}_{01}(\zeta)\right\} \approx \#\left\{j : p_j \in \tilde{S}_{01}(\zeta)\right\},$$

$$\#\left\{j \in \Delta_{10} \cup \Delta_{00} : p_j \in \widehat{S}(\zeta)\right\} \approx \#\left\{j \in \Delta_{10} \cup \Delta_{00} : p_j \in \tilde{S}_{10}(\zeta)\right\} \approx \#\left\{j : p_j \in \tilde{S}_{10}(\zeta)\right\},$$

$$\#\left\{j \in \Delta_{00} : p_j \in \widehat{S}(\zeta)\right\} \approx \#\left\{j \in \Delta_{00} : p_j \in \tilde{S}_{00}(\zeta)\right\} \approx \#\left\{j : p_j \in \tilde{S}_{00}(\zeta)\right\}.$$

The number of the selection $\#\left\{j : p_j \in \tilde{S}_{01}(\zeta)\right\} + \#\left\{j : p_j \in \tilde{S}_{10}(\zeta)\right\} - \#\left\{j : p_j \in \tilde{S}_{00}(\zeta)\right\} + 1$ can be considered as an overestimation of $\#\left\{j : j \in \Delta_{00} \cup \Delta_{01} \cup \Delta_{10}, j \in \widehat{\Delta}\right\}$, and $\widehat{\mathrm{FDP}}(\zeta)$ is given by

$$\widehat{\mathrm{FDP}}(\zeta) = \frac{\#\left\{j : p_j \in \tilde{S}_{01}(\zeta)\right\} + \#\left\{j : p_j \in \tilde{S}_{10}(\zeta)\right\} - \#\left\{j : p_j \in \tilde{S}_{00}(\zeta)\right\} + 1}{\#\{j : p_j \in \widehat{S}(\zeta)\} \vee 1}. \tag{19}$$

Then the data-driven cutoff $\zeta^\star$ can be determined as follows:

$$\zeta^\star = \sup\{\zeta > 0 : \widehat{\text{FDP}}(\zeta) \leq \alpha\}, \tag{20}$$

and the final selection is $\widehat{\Delta}_{\zeta^\star} = \left\{j : p_j \in \widehat{S}(\zeta^\star)\right\}$.

Finally, we summarize our proposed FDR control procedure in Algorithm 1.

---

**Algorithm 1** A data-driven algorithm for FDR control.

---

1: Calculate a pair of p-values $p_j = (p_{1j}, p_{2j})$ following model (2), where $j = 1, \ldots, J$.
2: Estimate the proportions of the composite null hypothesis $\hat{\pi}_{00}, \hat{\pi}_{01}, \hat{\pi}_{10}$.
3: Estimate the null densities $\hat{f}_{01}(p), \hat{f}_{10}(p)$ and the mixture density $\hat{f}(p)$ using the adaptation of the beta kernel estimator.
4: Estimate the $\widehat{\text{fdr}}(p)$ following (17).
5: For a nominal FDR level $\alpha \in (0, 1)$, select the mediators $\left\{j : p_j \in \widehat{S}(\zeta^\star)\right\}$ where $\widehat{S}(\zeta) = \{p : \widehat{\text{fdr}}(p) \leq \zeta\}$ and the cutoff $\zeta^\star$ is

$$\zeta^\star = \sup\left\{\zeta > 0 : \widehat{\text{FDP}}(\zeta) = \frac{\#\left\{j : p_j \in \tilde{S}_{01}(\zeta)\right\} + \#\left\{j : p_j \in \tilde{S}_{10}(\zeta)\right\} - \#\left\{j : p_j \in \tilde{S}_{00}(\zeta)\right\} + 1}{\#\{j : p_j \in \widehat{S}(\zeta)\} \vee 1} \leq \alpha\right\}.$$

---

**Remark 2.** *In a recent work of Deng et al. [13], a procedure called JM was introduced for detecting simultaneous signals across multiple independent experiments. The core idea behind JM is to partition the region of p-values into masked and unmasked areas, and then utilize p-values from each of these regions to estimate FDR and local FDR, respectively. By leveraging the partially revealed information from the unmasked area, JM updates the rejection region in the masked area iteratively until it reaches the desired FDR level. In contrast to the stepwise updates in the JM procedure, the AMDP does not require such iterative adjustment. By leveraging information from large-scale testing, AMDP can accurately estimate the local FDR. Additionally, motivated by the symmetric property of the composite null hypothesis, we proposed a data-driven algorithm to determine the optimal rejection region. As shown in (19), the number of the selection $\#\left\{j : p_j \in \tilde{S}_{01}(\zeta)\right\} + \#\left\{j : p_j \in \tilde{S}_{10}(\zeta)\right\} - \#\left\{j : p_j \in \tilde{S}_{00}(\zeta)\right\} + 1$, provides a less conservative estimation of $\#\left\{j : j \in \Delta_{00} \cup \Delta_{01} \cup \Delta_{10}, j \in \widehat{\Delta}\right\}$ than JM, which relies on conditional mirror conservation to estimate FDP in masked region.*

Theorem 2 below shows that for any nominal FDR level $\alpha \in (0, 1)$, both $\text{FDP}(\zeta^\star)$ and $\text{FDR}(\zeta^\star)$ are under control using Algorithm 1, as the sample size $n$ and the number of mediators $J$ tend to infinity.

**Theorem 2.** *Assume that*

*(i)* $\frac{1}{J} \sum_{j=1}^{J} \left|\widehat{\text{fdr}}(p_j) - \text{fdr}(p_j)\right| \xrightarrow{P} 0$ *as* $n, J \to \infty$;

*(ii) For* $\zeta \in (0, 1]$, $\mathbb{P}\left(\text{fdr}(p_j) \leq \zeta \mid j \in \Delta_{00} \cup \Delta_{01} \cup \Delta_{10}\right)$ *is continuous;*

*(iii) For any FDR level of* $\alpha \in (0, 1)$, *there exists a constant* $\zeta_\alpha \in (0, 1]$ *such that* $\mathbb{P}\left(\text{FDP}(\zeta_\alpha) \leq \alpha\right) \to 1$ *as* $J \to \infty$.

*When Assumptions 1-2 holds, we have*

$$\text{FDP}\left(\zeta^\star\right) \leq \alpha + o_p(1) \text{ and } \limsup_{n, J \to \infty} \text{FDR}\left(\zeta^\star\right) \leq \alpha.$$

Proofs of Theorems 1-2 are given in Appendix E.

**Remark 3.** *To demonstrate the effectiveness of our proposed method, we provide examples under two scenarios that highlight how the loss of associated information across tests can result in decreased power. We compare our method, AMDP, with the JS-mixture test[12] and the DACT test[30], which provides further evidence for the utility of AMDP. Due to space limitations, this section is postponed to the Appendix B.*

## 3 Simulation Study

In this section, we conduct a thorough set of simulations to assess the performance of our proposed method AMDP. For a comprehensive comparison, we evaluate two competing methods, the JS-mixture [12] and the DACT [30]. The DACT method consists of two variants: DACT (Efron) and

DACT (JC). The R implementations of JS-mixture and DACT can be found at `https://github.com/cran/HDMT` and `https://github.com/zhonghualiu/DACT`, respectively. The exposure $X$ is simulated from $\text{Ber}(0.5)$, then the mediator $M_j$ and the outcome $Y_j$ are generated as follows:

$$Y_j = \beta_j M_j + \epsilon_j, \epsilon_j \sim N(0,1),$$
$$M_j = \alpha_j X + e_j, e_j \sim N(0,1).$$

We respectively calculate the FDP and true discovery proportion (TDP) as follows:

$$\text{FDP} = \frac{\#\left\{j : j \in \Delta_0, j \in \widehat{\Delta}\right\}}{\#\{j \in \widehat{\Delta}\} \vee 1}, \quad \text{TDP} = \frac{\#\left\{j : j \notin \Delta_0, j \in \widehat{\Delta}\right\}}{\#\{j \notin \Delta_0\} \vee 1}. \tag{21}$$

where $\Delta_0$ is the index set of the composite null mediators, $\widehat{\Delta}$ is the index set of selected mediators. The FDR and power are measured by averaging FDP and TDP over 200 replications, respectively. Let $\tau$ be the mediation effect size parameter. We utilize the following six examples to conduct a comprehensive comparison of the FDR and power for the four procedures.

**Example 1.** We fix $(n, J) = (1000, 10000)$, $(\pi_{00}, \pi_{01}, \pi_{10}, \pi_{11}) = (0.85, 0.05, 0.05, 0.05)$, and vary $\tau$ from 0.6 to 1.2. Under $\text{H}_{00}$, $\alpha_j = 0$ and $\beta_j = 0$; under $\text{H}_{01}$, $\alpha_j = 0$ and $\beta_j = 0.3\tau$; under $\text{H}_{10}$, $\alpha_j = 0.2\tau$ and $\beta_j = 0$; under $\text{H}_{11}$, $\alpha_j = 0.2\tau$ and $\beta_j = 0.3\tau$;

**Example 2.** We fix $(n, J) = (1000, 10000)$, $(\pi_{00}, \pi_{01}, \pi_{10}, \pi_{11}) = (0.4, 0.2, 0.2, 0.2)$, and vary $\tau$ from 0.6 to 1.2. Under $\text{H}_{00}$, $\alpha_j = 0$ and $\beta_j = 0$; under $\text{H}_{01}$, $\alpha_j = 0$ and $\beta_j = 0.3\tau$; under $\text{H}_{10}$, $\alpha_j = 0.2\tau$ and $\beta_j = 0$; under $\text{H}_{11}$, $\alpha_j = 0.2\tau$ and $\beta_j = 0.3\tau$;

**Example 3.** We fix $(n, \tau) = (1000, 1)$, $(\pi_{00}, \pi_{01}, \pi_{10}, \pi_{11}) = (0.85, 0.05, 0.05, 0.05)$, and vary $J \in \{5000, 8000, 10000, 20000\}$. Under $\text{H}_{00}$, $\alpha_j = 0$ and $\beta_j = 0$; under $\text{H}_{01}$, $\alpha_j = 0$ and $\beta_j = 0.3\tau$; under $\text{H}_{10}$, $\alpha_j = 0.2\tau$ and $\beta_j = 0$; under $\text{H}_{11}$, $\alpha_j = 0.2\tau$ and $\beta_j = 0.3\tau$;

**Example 4.** We fix $(n, \tau) = (1000, 1)$, $(\pi_{00}, \pi_{01}, \pi_{10}, \pi_{11}) = (0.4, 0.2, 0.2, 0.2)$, and vary $J \in \{5000, 8000, 10000, 20000\}$. Under $\text{H}_{00}$, $\alpha_j = 0$ and $\beta_j = 0$; under $\text{H}_{01}$, $\alpha_j = 0$ and $\beta_j = 0.3\tau$; under $\text{H}_{10}$, $\alpha_j = 0.2\tau$ and $\beta_j = 0$; under $\text{H}_{11}$, $\alpha_j = 0.2\tau$ and $\beta_j = 0.3\tau$;

**Example 5.** We fix $(J, \tau) = (10000, 1)$, $(\pi_{00}, \pi_{01}, \pi_{10}, \pi_{11}) = (0.85, 0.05, 0.05, 0.05)$, and vary $n \in \{600, 800, 1000, 1200\}$. Under $\text{H}_{00}$, $\alpha_j = 0$ and $\beta_j = 0$; under $\text{H}_{01}$, $\alpha_j = 0$ and $\beta_j = 0.3\tau$; under $\text{H}_{10}$, $\alpha_j = 0.2\tau$ and $\beta_j = 0$; under $\text{H}_{11}$, $\alpha_j = 0.2\tau$ and $\beta_j = 0.3\tau$.

**Example 6.** We fix $(J, \tau) = (10000, 1)$, $(\pi_{00}, \pi_{01}, \pi_{10}, \pi_{11}) = (0.4, 0.2, 0.2, 0.2)$, and vary $n \in \{600, 800, 1000, 1200\}$. Under $\text{H}_{00}$, $\alpha_j = 0$ and $\beta_j = 0$; under $\text{H}_{01}$, $\alpha_j = 0$ and $\beta_j = 0.3\tau$; under $\text{H}_{10}$, $\alpha_j = 0.2\tau$ and $\beta_j = 0$; under $\text{H}_{11}$, $\alpha_j = 0.2\tau$ and $\beta_j = 0.3\tau$.

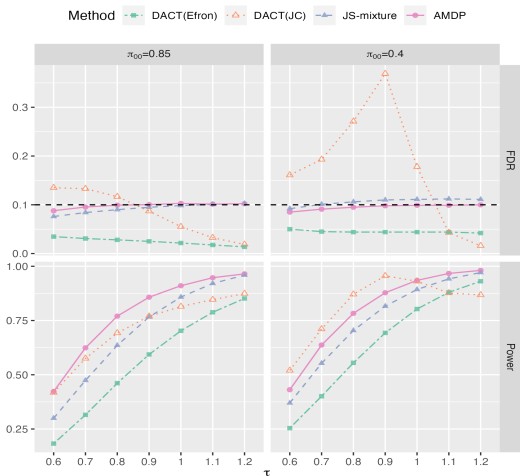

Figure A1: The average FDR and power performance of four methods under Examples 1-2. The sample size $n$ is 1000 and the number of mediators $J$ is 10000. The effect size parameter $\tau$ varies from 0.6 to 1.2. The nominal FDR level is 0.1.

To assess how the four methods are affected by effect size under sparse alternatives ($\pi_{11} = 0.05$) and dense alternatives ($\pi_{11} = 0.2$), we apply the four methods in Examples 1-2. The effect size, $\tau$, is varied from 0.6 to 1.2 in both examples. The results of estimated FDR and power are summarized in Figure A1. For sparse alternatives in Example 1, AMDP and JS-mixture maintain stable FDR control at the nominal level across various effect sizes. DACT (Efron) consistently controls the FDR but can be overly conservative, leading to potential under-identification of significant signals. Moreover, the FDR level of DACT (JC) exhibits inflation under weak effects. In terms of power analysis, AMDP emerges as the top performer, consistently outperforming the other three methods in Example 1. JS-mixture ranks second when the effect is strong, while DACT (Efron) lags behind due to its conservative behavior. For dense alternatives in Example 2, DACT (JC) fails to effectively control the FDR, leading to substantially higher FDR than the nominal level. While DACT (Efron) still exhibits overly conservative behavior. However, AMDP and JS-mixture maintain stable FDR levels across different effect sizes in Example 2, highlighting their robustness in controlling FDR. Regarding power analysis in Example 2, AMDP consistently outperforms the other methods, demonstrating its ability to handle scenarios with a substantial proportion of $H_{01}$ and $H_{10}$ while still achieving high power. JS-mixture ranks second in terms of power. It is noteworthy that in certain settings of Example 2, DACT (JC) may exhibit higher power than AMDP. Nevertheless, this higher power is often associated with severely inflated FDR levels. The conservation of DACT (Efron) results in lower power compared to the other three methods.

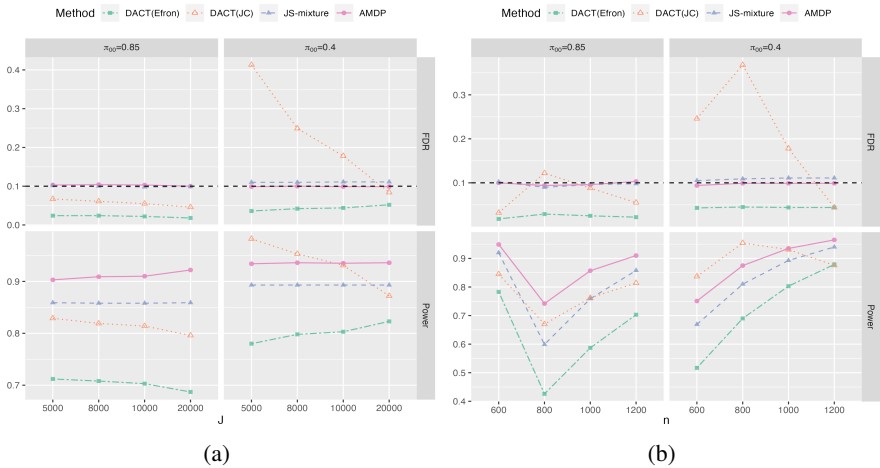

(a)            (b)

Figure A2: The average FDR and power performance of four methods under the targeted FDR level of 0.1. (a) The comparison under Examples 3-4 for varying the number of mediators $J \in \{5000, 8000, 10000, 20000\}$ with $n = 1000$, $\tau = 1$; (b) The comparison under Examples 5-6 for varying the sample size $n \in \{600, 800, 1000, 1200\}$ with $J = 10000$, $\tau = 1$.

Next, we move on to investigate whether the four methods are sensitive to changes in the large mediator size $J$ under both sparse and dense alternative scenarios. Panel (a) of Figure A2 displays the FDR and power performance of the four methods under Examples 3-4. In the sparse alternatives scenario of Example 3, AMDP and JS-mixture demonstrate remarkable stability in controlling FDR at the nominal level across different values of $J$. DACT (Efron) exhibits conservative FDR control. While DACT (JC) is less conservative than DACT (Efron), it remains underpowered. In terms of power analysis, AMDP and JS-mixture are the leading methods. DACT (JC) demonstrates higher power when $J$ is not very large, but its power decreases as $J$ grows. DACT (Efron) consistently displays lower power in all settings due to its conservative behavior. In the dense alternative scenario of Example 4, DACT (JC) suffers from inflated FDR. In contrast, the FDRs of the other three methods are under control with varying mediator sizes, though DACT (Efron) continues to be overly conservative. Moreover, AMDP consistently delivers the highest power among all methods in the dense alternative scenario. We note that JS-mixture performs competitively in terms of power. On the other hand, the power of DACT (JC) decreases with the growth of $J$, which raises concerns about its ability to detect true positives accurately in scenarios with larger mediator sizes. DACT (Efron) consistently displays lower power in all settings.

In Examples 5-6, we explore the influence of sample size on FDR and power performance under sparse and dense alternatives, respectively. The results of these analyses are presented in panel (b) of

Figure A2. In Example 5, where a small proportion of alternative hypotheses is considered, AMDP and JS-mixture stand out as more accurate and stable in controlling the FDR among all methods. When the sample size is small, DACT (JC) exhibits a slightly higher FDR compared to AMDP and JS-mixture. DACT (Efron) remains overly conservative. The power of the four approaches initially decreases and then increases with the growth of $n$. Specially, for all four methods, the lowest power is achieved at $n = 800$ among all the tested sample sizes. Moreover, AMDP consistently delivers reasonably higher power compared to the other three methods. In the dense alternatives of Example 6, DACT (JC) encounters challenges in maintaining FDR control, particularly when $n$ is small. In contrast, AMDP, JS-mixture, and DACT (Efron) effectively control the FDR across different settings. Regarding power performance, AMDP, JS-mixture, and DACT (Efron) demonstrate a consistent increase in power as $n$ grows. In some settings, DACT (JC) appears to perform better than AMDP. Nevertheless, this seemingly higher power of DACT (JC) is a result of the severely inflated FDR levels. We note that the consistent superiority of AMDP in both FDR control and power, as observed in Examples 5 and 6, aligns with the theoretical results presented in Theorem 1.

## 4   Data Analysis

Prostate cancer is a prevalent disease among men, with a multifactorial etiology involving genetic, environmental, and lifestyle factors. There is a growing recognition that DNA methylation plays an important role in regulating gene expression [44]. Additionally, the number of GWASs-identified risk SNP that influence DNA methylation levels in prostate cancer has reached a total of 167 [6]. Despite significant progress in understanding the role of DNA methylation in gene expression regulation and identifying prostate cancer risk SNPs, further research is strongly encouraged to uncover the specific CpG sites that contribute to the regulatory effects of risk SNPs on their target genes.

We apply our proposed AMDP, JS-mixture [12], and DACT [30], including DACT (Efron) and DACT (JC), to analyze the TCGA prostate cancer dataset. The dataset is freely available at `https://portal.gdc.cancer.gov`. Our analysis focuses on 495 primary prostate tumor samples with information on 147 prostate cancer risk SNPs, DNA methylation, and gene expression. In total, we consider 69,602 CpG methylation probes ($M$) as potential mediators. The risk SNPs are the exposure variable ($X$), and gene expression is the outcome variable of interest ($Y$). The primary objective of our analysis is to explore the potential causal role of CpG methylation in the association between prostate cancer risk SNPs and gene expression. We estimate the null proportions as $\hat{\pi}_{00} = 0.52$, $\hat{\pi}_{10} = 0.03$, and $\hat{\pi}_{01} = 0.42$, respectively. Figure A2 of the Appendix C displays the number of significant triplets ($X - M - Y$) detected by AMDP, JS-mixture, DACT (Efron), and DACT (JC) at different nominal FDR levels $\alpha$ ranging from 0.01 to 0.1. It can be seen that, in the majority of cases, AMDP outperforms the other three methods by identifying more triplets at the same FDR level. On average, the discoveries made by AMDP are approximately 20.2% higher than those of JS-mixture, 86.6% higher than those of DACT (Efron), and 42.6% higher than those of DACT (JC), across the range of $\alpha$ values from 0.01 to 0.1. This substantial improvement in performance highlights the effectiveness of AMDP in identifying non-zero mediation effects in the prostate cancer dataset.

Additional results of Section 4 are postponed to the Appendix C.

## 5   Discussion

In this paper, we develop a novel adaptive mediation detection procedure (AMDP) to identify significant mediators in high-dimensional mediation analysis. The novel approach determines the optimal ranking for hypotheses, and then employs a data-driven strategy to select the threshold for mediator identification. We demonstrate the effectiveness of our proposed method through theoretical analysis and simulation results. There is a potential avenue for future research. We discuss the mediation effect based on the marginal model in this paper, where the p-values are independent. How to further study relevant mediators from two aspects of theory and application is an interesting topic.

**Limitation** Our approach effectively handles high-dimensional mediators but may not perform optimally when confronted with low-dimensional mediators. This distinction is attributed to the nature of our method, wherein the two-dimensional p-values linked to each exposure-mediator-outcome relationship effectively serve as "samples" for the estimation of local FDR and FDP. Consequently, the reduction in dimensionality can lead to less precise estimates of local FDR and FDP.

## Acknowledgements

The authors thank the editor and the anonymous reviewers for their constructive suggestions that significantly improved an early manuscript. The research described herewith was supported by a grant from the National Key R&D Program of China (2022YFA1003803), a grant from the National Social Science Foundation of China (21BTJ048), a grant from the National Scientific Foundation of China (12371276, 12131006), Basic Science Research Foundation of Shaanxi (22JSY038) and the Zhongying Young Scholar Program.

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

# AMDP: An Adaptive Detection Procedure for False Discovery Rate Control in High-Dimensional Mediation Analysis

# Appendix

## A    Selection of an appropriate $\lambda$

In Section 2.2, we estimate $\hat{\pi}_{00}(\lambda)$, $\hat{\pi}_{10}(\lambda)$, and $\hat{\pi}_{01}(\lambda)$ using the method proposed by Storey et al. [41] with a fixed parameter $\lambda$. The estimators are calculated in (10)-(11). Theoretical considerations suggest that as $\lambda$ approaches 1, the estimators of the composite null hypothesis become more accurate asymptotically. However, in finite samples scenarios, with a larger value of $\lambda$, the chance of these null p-values falling within $(\lambda, 1]$ gets smaller, resulting in less accurate estimates. Conversely, when $\lambda$ becomes smaller, the bias of the null estimators increases while the variance decreases [41]. Consequently, there exists an inherent bias-variance trade-off in the selection of $\lambda$.

To strike a reasonable balance between bias and variance, we aim to determine $\lambda$ by minimizing the mean-squared error (MSE) of the estimators. The MSE is defined as $E[\{\hat{\pi}_{00}(\lambda) - \pi_{00}\}^2 + \{\hat{\pi}_{10}(\lambda) - \pi_{10}\}^2 + \{\hat{\pi}_{01}(\lambda) - \pi_{01}\}^2]$. For achieving this goal, we consider a range of cutpoints for $\lambda$ (e.g., $\lambda = 0.1, 0.2, \ldots, 0.9$) and calculate the MSE for each value of $\lambda$. As highlighted by Barfield et al. [3], a substantial proportion of null hypotheses may exhibit both $\alpha = 0$ and $\beta = 0$ in a genome-wide study involving high-dimensional mediation hypotheses. To investigate the choice of $\lambda$ in such scenarios, we consider the following settings, as shown in Table A1.

Table A1: The composite null proportions under different scenarios.

| Hypothesis Configuration | $\pi_{00}$ | $\pi_{10}$ | $\pi_{01}$ | $\pi_{11}$ |
|---|---|---|---|---|
| Scenario 1 | 0.2 | 0.3 | 0.3 | 0.2 |
| Scenario 2 | 0.4 | 0.2 | 0.2 | 0.2 |
| Scenario 3 | 0.5 | 0.2 | 0.2 | 0.1 |
| Scenario 4 | 0.6 | 0.15 | 0.15 | 0.1 |
| Scenario 5 | 0.75 | 0.1 | 0.1 | 0.05 |
| Scenario 6 | 0.85 | 0.05 | 0.05 | 0.05 |

Table A2 shows means and MSE of the estimated null proportions under the six scenarios with $n$=1000, $J$=10000, $\alpha_j$= 0.2, and $\beta_j$=0.3. In each scenario, we identified the top three smallest MSEs among the estimated null proportions. Notably, we observed that the optimal value of $\lambda$ varied across the different scenarios. However, it is noteworthy that $\lambda = 0.5$ consistently appeared among the top three MSE values in all the simulated scenarios. As mentioned earlier, smaller values of $\lambda$ tend to result in larger biases of the null estimate, while excessively large values of $\lambda$ may yield inaccurate estimates in finite sample scenarios. Therefore, the consistent appearance of $\lambda = 0.5$ among the top-performing MSE values suggests that it provides a reasonable trade-off between bias and variance, leading to accurate estimates across a wide range of scenarios. Considering its stability and computational efficiency, we believe that $\lambda = 0.5$ is a suitable choice for estimating the null proportions in high-dimensional mediation analysis.

## B    Comparison with existing methods

In this section, we demonstrate the loss of information during the ranking step can result in decreased statistical power, despite controlling the FDR at the desired level.

Table A2: The performance of the estimated proportions of the composite null hypothesis (mean and MSE) under the six scenarios. The tuning parameter $\lambda$ varies from 0.1 to 0.9.

| Scenario 1 | $\hat{\pi}_{00}$ | $\hat{\pi}_{10}$ | $\hat{\pi}_{01}$ | MSE | Scenario 2 | $\hat{\pi}_{00}$ | $\hat{\pi}_{10}$ | $\hat{\pi}_{01}$ | MSE |
|---|---|---|---|---|---|---|---|---|---|
| $\lambda$=0.1 | 0.222 | 0.278 | 0.314 | 1.17e-3 | $\lambda$=0.1 | 0.415 | 0.185 | 0.215 | 6.41e-4 |
| $\lambda$=0.2 | 0.211 | 0.289 | 0.308 | 3.10e-4 | $\lambda$=0.2 | 0.408 | 0.192 | 0.208 | 1.88e-4 |
| $\lambda$=0.3 | 0.207 | 0.293 | 0.306 | 1.20e-4 | $\lambda$=0.3 | 0.406 | 0.194 | 0.205 | 9.14e-5 |
| $\lambda$=0.4 | 0.205 | 0.295 | 0.305 | 6.69e-5 | $\lambda$=0.4 | 0.404 | 0.196 | 0.204 | **4.61e-5** |
| $\lambda$=0.5 | 0.203 | 0.296 | 0.303 | **3.03e-5** | $\lambda$=0.5 | 0.402 | 0.196 | 0.203 | **2.56e-5** |
| $\lambda$=0.6 | 0.203 | 0.296 | 0.303 | 3.25e-5 | $\lambda$=0.6 | 0.403 | 0.197 | 0.202 | **2.46e-5** |
| $\lambda$=0.7 | 0.202 | 0.297 | 0.302 | **1.96e-5** | $\lambda$=0.7 | 0.406 | 0.194 | 0.198 | 7.62e-5 |
| $\lambda$=0.8 | 0.201 | 0.299 | 0.302 | **5.94e-6** | $\lambda$=0.8 | 0.405 | 0.195 | 0.199 | 5.96e-5 |
| $\lambda$=0.9 | 0.206 | 0.295 | 0.296 | 8.54e-5 | $\lambda$=0.9 | 0.405 | 0.192 | 0.198 | 8.69e-5 |
| Scenario 3 | $\hat{\pi}_{00}$ | $\hat{\pi}_{10}$ | $\hat{\pi}_{01}$ | MSE | Scenario 4 | $\hat{\pi}_{00}$ | $\hat{\pi}_{10}$ | $\hat{\pi}_{01}$ | MSE |
| $\lambda$=0.1 | 0.515 | 0.185 | 0.207 | 4.84e-4 | $\lambda$=0.1 | 0.611 | 0.139 | 0.157 | 2.94e-4 |
| $\lambda$=0.2 | 0.507 | 0.192 | 0.204 | 1.38e-4 | $\lambda$=0.2 | 0.606 | 0.144 | 0.154 | 8.07e-5 |
| $\lambda$=0.3 | 0.505 | 0.194 | 0.203 | 6.82e-5 | $\lambda$=0.3 | 0.604 | 0.146 | 0.152 | 3.79e-5 |
| $\lambda$=0.4 | 0.504 | 0.196 | 0.201 | 3.81e-5 | $\lambda$=0.4 | 0.602 | 0.148 | 0.152 | **1.18e-5** |
| $\lambda$=0.5 | 0.503 | 0.197 | 0.201 | **1.91e-5** | $\lambda$=0.5 | 0.602 | 0.148 | 0.151 | **7.87e-6** |
| $\lambda$=0.6 | 0.502 | 0.196 | 0.202 | 2.24e-5 | $\lambda$=0.6 | 0.603 | 0.147 | 0.150 | 1.79e-5 |
| $\lambda$=0.7 | 0.503 | 0.195 | 0.199 | 3.15e-5 | $\lambda$=0.7 | 0.604 | 0.148 | 0.148 | 2.12e-5 |
| $\lambda$=0.8 | 0.501 | 0.197 | 0.202 | **1.36e-5** | $\lambda$=0.8 | 0.604 | 0.147 | 0.147 | 2.92e-5 |
| $\lambda$=0.9 | 0.501 | 0.198 | 0.201 | **6.69e-6** | $\lambda$=0.9 | 0.602 | 0.150 | 0.150 | **3.19e-6** |
| Scenario 5 | $\hat{\pi}_{00}$ | $\hat{\pi}_{10}$ | $\hat{\pi}_{01}$ | MSE | Scenario 6 | $\hat{\pi}_{00}$ | $\hat{\pi}_{10}$ | $\hat{\pi}_{01}$ | MSE |
| $\lambda$=0.1 | 0.757 | 0.093 | 0.104 | 1.22e-4 | $\lambda$=0.1 | 0.854 | 0.046 | 0.053 | 4.12e-5 |
| $\lambda$=0.2 | 0.754 | 0.096 | 0.103 | 3.69e-5 | $\lambda$=0.2 | 0.853 | 0.048 | 0.052 | 1.25e-5 |
| $\lambda$=0.3 | 0.753 | 0.097 | 0.102 | 2.22e-5 | $\lambda$=0.3 | 0.853 | 0.048 | 0.050 | 1.19e-5 |
| $\lambda$=0.4 | 0.751 | 0.099 | 0.102 | **6.94e-6** | $\lambda$=0.4 | 0.852 | 0.049 | 0.050 | **3.79e-6** |
| $\lambda$=0.5 | 0.752 | 0.099 | 0.101 | **3.38e-6** | $\lambda$=0.5 | 0.850 | 0.049 | 0.051 | **9.69e-7** |
| $\lambda$=0.6 | 0.754 | 0.097 | 0.099 | 2.35e-5 | $\lambda$=0.6 | 0.851 | 0.050 | 0.051 | **1.40e-6** |
| $\lambda$=0.7 | 0.755 | 0.096 | 0.098 | 4.53e-5 | $\lambda$=0.7 | 0.849 | 0.052 | 0.052 | 1.16e-5 |
| $\lambda$=0.8 | 0.751 | 0.098 | 0.101 | **5.44e-6** | $\lambda$=0.8 | 0.848 | 0.054 | 0.056 | 6.07e-5 |
| $\lambda$=0.9 | 0.765 | 0.091 | 0.090 | 4.09e-4 | $\lambda$=0.9 | 0.857 | 0.056 | 0.055 | 1.11e-4 |

To conduct this investigation, we consider three different approaches for comparison: our proposed AMDP, along with two existing methods, the JS-mixture [12] and the DACT [30]. During the selection step, it is assumed that the information about the proportions of the composite null hypothesis and the distributions of p-values under alternatives are known. This provided knowledge allows for effectively controlling the FDR of all three procedures at the predefined level of $\alpha$ in the selection step. With the FDR under control, we then proceed to investigate how different ranking strategies impact the power performance of the three methods. To achieve this, we compare the rejection regions of each method under various scenarios. Formally, the ranking statistic for each method is as follows:

$$\delta^{JS-mixture} = p_{max},$$
$$\delta^{DACT} = \omega_1 p^{(1)} + \omega_2 p^{(2)} + \omega_3 p_{max},$$
$$\delta^{AMDP} = \mathrm{fdr}(p),$$

where $p_{max} = p^{(1)} \vee p^{(2)}$, $\vee$ denotes the maximum of the two p-values. $\omega_1$, $\omega_2$ and $\omega_3$ are normalized relative proportions of the composite null. We consider two scenarios to compare the ranking statistic of the three methods:

**Scenario 1**    Balanced null proportions of $H_{01}$ and $H_{10}$:

$$f(p^{(1)}, p^{(2)}) = 0.49 + 0.21 \times 0.6 p^{(1)^{-0.4}} + 0.21 \times 0.3 p^{(2)^{-0.7}} + 0.09 \times 0.18 p^{(1)^{-0.4}} p^{(2)^{-0.7}},$$

where the density functions of p-values under alternatives are $f(p^{(1)} \mid H_{10}) \sim Beta(0.6, 1)$ and $f(p^{(2)} \mid H_{01}) \sim Beta(0.3, 1)$. The proportions of composite hypothesis are $\pi_{00} = 0.49$, $\pi_{01} = \pi_{10} = 0.21$, and $\pi_{11} = 0.09$.

**Scenario 2**    Unbalanced null proportions of $H_{01}$ and $H_{10}$:

$$f(p^{(1)}, p^{(2)}) = 0.4 + 0.1 \times 0.4 p^{(1)^{-0.6}} + 0.4 \times 0.6 p^{(2)^{-0.4}} + 0.1 \times 0.24 p^{(1)^{-0.6}} p^{(2)^{-0.4}},$$

where the density functions of p-values under alternatives are $f(p^{(1)} \mid H_{10}) \sim Beta(0.4, 1)$ and $f(p^{(2)} \mid H_{01}) \sim Beta(0.6, 1)$. The proportions of composite hypothesis are $\pi_{00} = 0.4$, $\pi_{01} = 0.4$, $\pi_{10} = 0.1$, and $\pi_{11} = 0.1$.

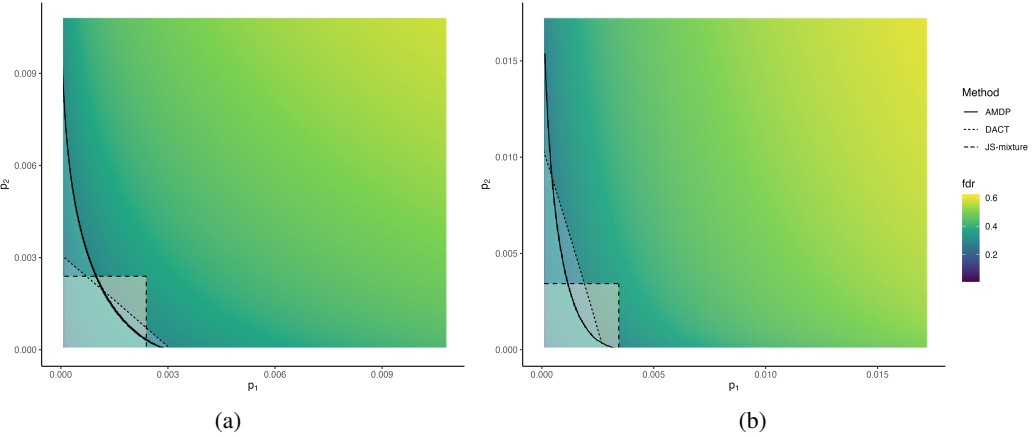

Figure A3: (a) The rejection regions of the three methods under Scenario 1; (b) The rejection regions of the three methods under Scenario 2.

Theorem 1 has proved that the ranking statistic of AMDP is optimal among those methods that effectively control the FDR at the nominal level. Therefore, we refer to the rejection region of AMDP as the oracle rejection region in the sense that it achieves the highest power under FDR control. In Scenarios 1-2, we compare the rejection regions of JS-mixture and DACT with the oracle rejection region when the FDR level can be precisely controlled to the specified level. This comparison provides deep insights into the impact of information loss during the ranking step on power.

Figure A3 visually presents the local FDR for the four-group model (5) in both Scenarios 1-2, as well as the rejection region for the three procedures. The color intensity in the figure represents the level of local FDR, with darker colors indicating lower local FDR, thus the corresponding hypothesis is more likely to be rejected. While the rejection regions of all three methods are all located in areas with lower local FDR, we emphasize that the AMDP is superior since it simultaneously considers information about proportions of the composite null hypothesis and the distributions of p-values under alternatives, leading to more accurate and reliable selection of mediators. In contrast, the other two methods only take into account partial information during the ranking step, which results in decreased power. The insensitivity of the JS-mixture method to changes in proportions and distributions is noteworthy. As described in panels (a)-(b), the shape of its rejection region remains square regardless of the scenarios. On the other hand, the DACT method only considers partial information about proportions of the null, since its rejection domain remains symmetrical under Scenario 1 (Balanced proportion of $H_{01}$ and $H_{10}$), and shifts towards the larger distribution side under Scenario 2 (Unbalanced proportion of $H_{01}$ and $H_{10}$). In contrast, the AMDP method fully captures all relevant information above, as the oracle rejection region is sensitive to the change in both proportions and distributions. This allows the AMDP method to adapt and adjust its rejection region accordingly, making it more effective in identifying significant signals.

## C  Additional results of Section 4

In this section, we demonstrate additional results related to the data analysis of the prostate cancer dataset in Section 4, including Table A3 and Figures A4-A5.

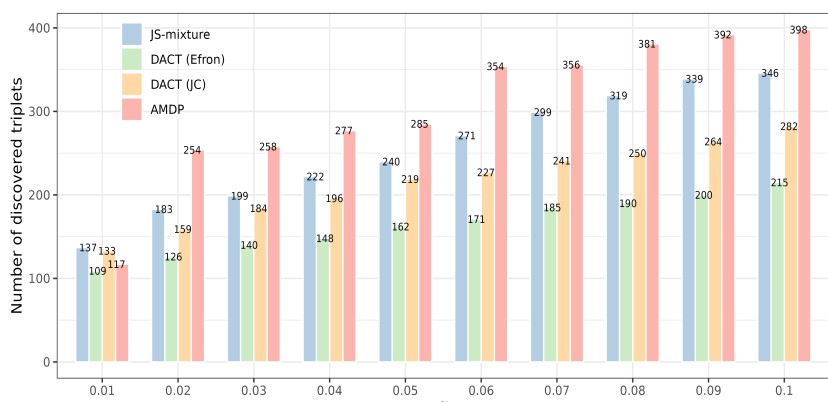

Figure A4: The number of triplets discovered by four methods. The nominal FDR level $\alpha$ varies from $0.01$ to $0.1$. The blue, green, orange, and pink bars represent the numbers of triplets identified by JS-mixture, DACT (Efron), DACT (JC), and AMDP, respectively.

Figure A5 provides an overview observation of the prostate cancer dataset as well as the rejection regions of JS-mixture, DACT (Efron), DACT (JC), and AMDP. Panel (a) shows the dispersion of p-values. However, the high density of the p-values makes it difficult to observe carefully. Thus, we depict the details of the TCGA dataset in different aspects in panels (b), (c), and (d), respectively, for providing a clearer insight. In panels (e)-(h), and (i)-(l), we compare the rejection regions of four methods: JS-mixture, DACT (Efron), DACT (JC), and AMDP at FDR levels of 0.05 and 0.1, respectively.

From panels (b)-(d), it can be seen that the distribution of p-values is influenced by information related to the composite null hypothesis. In panel (b), there is a slightly denser concentration of p-values near the $p^{(1)} = 0$ axis. This occurrence can be attributed to the presence of $\hat{\pi}_{10} = 0.03$, as mentioned earlier. On the other hand, panel (c) exhibits a notable concentration of p-values near the $p^{(2)} = 0$ axis. This pattern is influenced by the presence of a significant number of cases falling under $H_{01}$, which affects the distribution of p-values. As a result, we observe an accumulation of p-values near the $p^{(2)} = 0$ axis in the plot. Panel (d) demonstrates a seemingly uniform distribution of p-values. This uniformity can be attributed to the theoretical expectation that only p-values under

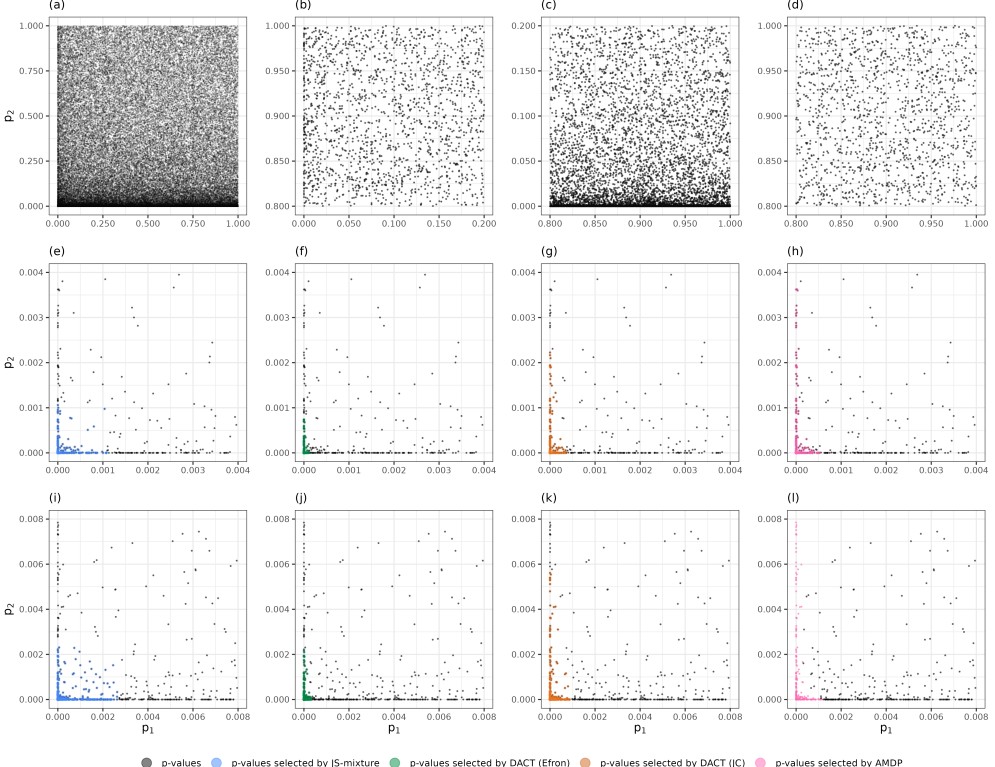

Figure A5: (a) An overview of p-values obtained from TCGA prostate cancer dataset; (b) The dispersion of p-values in the region $[0, 0.2] \times [0.8, 1]$; (c) The dispersion of p-values in the region $[0.8, 1] \times [0, 0.2]$;(d) The dispersion of p-values in the region $[0.8, 1] \times [0.8, 1]$; (e)-(h) The rejection domains of JS-mixture, DACT (Efron), DACT (JC), and AMDP at the targeted FDR level 0.05, respectively; (i)-(l) The rejection domains of JS-mixture, DACT (Efron), DACT (JC), and AMDP at the targeted FDR level 0.1, respectively; The black dots represent the p-values of all triplets. The blue, green, orange, and pink dots represent p-values of triplets identified by JS-mixture, DACT (Efron), DACT (JC), and AMDP, respectively.

$H_{00}$ exist in the region $[0.8, 1] \times [0.8, 1]$. At the FDR level of 0.05, the rejection region of JS-mixture in panel (e) corresponds to a square shape. However, this symmetric shape does not reflect any information related to the distribution of p-values or the proportions of the composite null. In contrast, DACT (Efron) considers the proportion of null hypotheses and demonstrates a preference for rejecting fewer hypotheses with p-values close to $p^{(2)} = 0$ to minimize false discoveries, as shown in panel (f). However, the number of triplets identified by DACT (Efron) is the least among all methods, resulting an overly conservative behavior. The conservatism observed in DACT (Efron) is alleviated by DACT (JC), as panel (g) reveals that DACT (JC) identifies more significant triplets compared to DACT (Efron). DACT (JC) offers a more efficient approach by adjusting the threshold of the rejection region to achieve a higher sensitivity. In panel (h), we observe that the rejection region of AMDP is adaptive. AMDP estimates the number of false discoveries based on symmetric regions of the rejection region, allowing for more effective and accurate control of false discoveries, and well-calibrated adjustments to the rejection region. AMDP strikes a better performance on detecting significant triplets among all procedures. Next, we turn to investigate the rejection regions of these four methods at the FDR level of 0.1. In panel (i), it is observed that the rejection region of JS-mixture remains insensitive to changes in FDR levels, maintaining its square shape. As shown in panels (j)-(k), both DACT (Efron) and DACT (JC) exhibit increased identification of triplets compared to the FDR level of 0.05. Nevertheless, they still appear to be somewhat underpowered in efficiently detecting significant triplets. In contrast, AMDP outperforms all the other methods at the same FDR level, as demonstrated in panel (l). By leveraging information on the proportions of null

and calibrating its rejection region dynamically, AMDP achieves better power to identify significant triplets.

Table A3: Top ten triplets identified by AMDP at the FDR level 0.1.

| SNP ID | CpG Name | Gene | Chromosome | $p_1$ | $p_2$ |
|---|---|---|---|---|---|
| rs12653946 | cg00626856 | IRX4 | 5 | 6.60e-56 | 2.66e-20 |
| rs12653946 | cg03587843 | IRX4 | 5 | 1.95e-51 | 1.03e-19 |
| rs12653946 | cg06161964 | IRX4 | 5 | 1.99e-53 | 2.02e-22 |
| rs12653946 | cg09672187 | IRX4 | 5 | 4.01e-65 | 2.13e-33 |
| rs12653946 | cg11279838 | IRX4 | 5 | 3.97e-64 | 3.61e-27 |
| rs12653946 | cg14051264 | IRX4 | 5 | 7.62e-67 | 8.86e-26 |
| rs12653946 | cg26195178 | IRX4 | 5 | 2.52e-61 | 2.43e-26 |
| rs5945619 | cg16065628 | NUDT11 | X | 6.75e-32 | 2.79e-42 |
| rs1933488 | cg23651356 | RGS17 | 6 | 7.25e-20 | 2.81e-16 |
| rs12653946 | cg14823763 | IRX4 | 5 | 8.13e-47 | 1.27e-15 |

The top ten triplets identified by AMDP are summaried in Table A3. These ten triplets consist of ten CpG sites and three genes. The CpG sites involved in these triplets are located in close proximity to the transcription starting sites, and their DNA methylation level are closely related to the expression of the corresponding genes [12]. Among the identified triplets, the three genes, IRX4, NUDT11, and RGS17, have been shown to be associated with altered CpG methylation. IRX4 is a causative gene of the prostate cancer susceptibility locus [45]. The corresponding SNP rs12653946, a variant previously confirmed to be associated with prostate cancer, is significantly associated with IRX4 expression [6]. The increased expression of NUDT11 has been confirmed to be associated with the risk variant rs5945619 [22, 29]. RGS17 is a commonly induced gene in prostate tumors, and has been found crucial for the maintenance of the proliferative potential of tumor cells [8].

# D  Discussions on the parameter choice and assumptions

## D.1  Parameter choice

In Figures 1-2 in Section 3, we assess how the four methods (JS-mixture, DACT (Efron), DACT (JC), and AMDP) are influenced by effect size, the large mediator size and sample size. To ensure the realism of our experiments, we carefully selected our simulation parameters. Motivated from several real-world datasets including the TCGA lung cancer cohort dataset [47], the Multi-Ethnic Study of Atherosclerosis [15], and the TCGA prostate cancer dataset [12], we adopt similar parameter settings as those used in [12] to construct the simulation examples in Section 3.

Regarding the choice of nominal FDR level, we initially used an FDR level of 0.1, which is a widely accepted standard in the field [12, 34]. Another common FDR level is 0.05 [23, 39]. To provide a comprehensive analysis, we conducted experiments at the FDR threshold of 0.05 across a wide range of sample sizes (200, 500, 1000, and 5000). We present the experimental results under sparse alternatives scenario and dense alternatives scenario in Tables A4-A5. It's noteworthy that the results are similar with those obtained using the FDR level of 0.1 in Section 3.

## D.2  Discussion on Assumptions 1-2

Our method extracts a pair $(p^{(1)}, p^{(2)})$ for each exposure-mediator-outcome relationship and employs these pairs to estimate the FDP on a two-dimensional plane $[0, 1] \times [0, 1]$. The theoretical basis supporting FDP estimation is the assumption that p-values are uniformly distributed under the null hypothesis, which is a widely recognized principle [7, 24]. Due to the presence of a composite null hypothesis in the mediation effect, we elaborate on Assumptions 1-2 to illustrate the properties of the p-value distribution under composite null hypothesis.

For Assumption 1, under $H_{00}$, both $p_{1j}$ and $p_{2j}$ obey the uniform distribution, resulting in $(p_{1j}, p_{2j})$ also following the uniform distribution on the two-dimensional plane $[0, 1] \times [0, 1]$. Consequently, the

Table A4: The FDR and power performance of the four methods with effect size $\alpha_j$=0.2, $\beta_j$=0.3 under sparse alternatives scenario. The nominal FDR level is 0.05, and the number of mediators is 15000.

|  | Method | FDR | Power |
|---|---|---|---|
| n=200 | AMDP | 0.0530 | 0.0474 |
|  | JS-mixture | 0.0284 | 0.0187 |
|  | DACT (Efron) | 0.0091 | 0.0055 |
|  | DACT (JC) | 0.0859 | 0.0960 |
| n=500 | AMDP | 0.0454 | 0.4665 |
|  | JS-mixture | 0.0438 | 0.3714 |
|  | DACT (Efron) | 0.0140 | 0.2254 |
|  | DACT (JC) | 0.0793 | 0.4960 |
| n=1000 | AMDP | 0.0488 | 0.8698 |
|  | JS-mixture | 0.0500 | 0.7931 |
|  | DACT (Efron) | 0.0114 | 0.6101 |
|  | DACT (JC) | 0.0299 | 0.7460 |
| n=5000 | AMDP | 0.0498 | 0.9999 |
|  | JS-mixture | 0.0529 | 1 |
|  | DACT (Efron) | 8.00e-05 | 0.9986 |
|  | DACT (JC) | 0.0935 | 1 |

Table A5: The FDR and power performance of the four methods with effect size $\alpha_j$=0.2, $\beta_j = 0.3$ under dense alternatives scenario. The nominal FDR level is 0.05, and the number of mediators is 15000.

|  | Method | FDR | Power |
|---|---|---|---|
| n=200 | AMDP | 0.0471 | 0.0331 |
|  | JS-mixture | 0.0348 | 0.0206 |
|  | DACT (Efron) | 0.0449 | 0.0379 |
|  | DACT (JC) | 0.1156 | 0.1673 |
| n=500 | AMDP | 0.0460 | 0.4768 |
|  | JS-mixture | 0.0504 | 0.4168 |
|  | DACT (Efron) | 0.0333 | 0.3488 |
|  | DACT (JC) | 0.1640 | 0.6839 |
| n=1000 | AMDP | 0.0487 | 0.8734 |
|  | JS-mixture | 0.0541 | 0.8208 |
|  | DACT (Efron) | 0.0315 | 0.7636 |
|  | DACT (JC) | 0.0818 | 0.8674 |
| n=5000 | AMDP | 0.0501 | 1 |
|  | JS-mixture | 0.0544 | 1 |
|  | DACT (Efron) | 0.0116 | 1 |
|  | DACT (JC) | 0.1085 | 1 |

sampling distribution of $(p_{1j}, p_{2j})$ is symmetrical around $p^{(1)} = 0.5$ and $p^{(2)} = 0.5$. Under $H_{01}$, $p_{1j}$ still obeys the uniform distribution, but $p_{2j}$ does not, leading to $(p_{1j}, p_{2j})$ being only symmetrical about $p^{(1)} = 0.5$ on $[0, 1] \times [0, 1]$. Similarly, under $H_{10}$, $p_{2j}$ obeys the uniform distribution, but $p_{1j}$ does not, resulting in $(p_{1j}, p_{2j})$ being only symmetrical about $p^{(2)} = 0.5$ on $[0, 1] \times [0, 1]$. It is essential to emphasize that Assumption 1 specifically applies to the null mediators.

For Assumption 2, a non-null p-value theoretically lies within $[0, 0.5)$. Therefore, as the sample size $n$ tends to infinity, the probability of p-values under alternative hypotheses falling within $[0.5, 1]$ approaches zero. For example, as $n$ goes to infinity, p-values under $H_{11}$ and $H_{10}$ are not expected to fall within the region $\tilde{D}_{01} = [0.5, 1] \times [0, 0.5)$ because non-null $p_{1j}$ not lies within $[0.5, 1]$ theoretically, therefore the region $\tilde{D}_{01}$ only contains p-values under $H_{00}$ and $H_{01}$. Similarly, the region $\tilde{D}_{10}$ theoretically only includes p-values under $H_{00}$ and $H_{10}$.

# E  Proofs

## E.1  Proof of Theorem 1

For any rejection region $S \in [0, 1]^2$, the global FDR in mediation analysis is defined as follows
$$\text{gFDR}(S) = \mathbb{P}(H_{00} \cup H_{01} \cup H_{10} = 1 \mid p_j \in S)$$

$$= \frac{\pi_{00}\mathbb{P}(p_j \in S \mid H_{00} = 1) + \pi_{01}\mathbb{P}(p_j \in S \mid H_{01} = 1) + \pi_{10}\mathbb{P}(p_j \in S \mid H_{10} = 1)}{\pi_{00}\mathbb{P}(p_j \in S \mid H_{00} = 1) + \pi_{01}\mathbb{P}(p_j \in S \mid H_{01} = 1) + \pi_{10}\mathbb{P}(p_j \in S \mid H_{10} = 1) + \pi_{11}\mathbb{P}(p_j \in S \mid H_{11} = 1)}$$

$$= \frac{\pi_{00}\int_S f_{00}(p)dp + \pi_{01}\int_S f_{01}(p)dp + \pi_{10}\int_S f_{10}(p)dp}{\pi_{00}\int_S f_{00}(p)dp + \pi_{01}\int_S f_{01}(p)dp + \pi_{10}\int_S f_{10}(p)dp + \pi_{11}\int_S f_{11}(p)dp}.$$
(A.1)

We introduce some notations
$$D_{00}(S) = \int_S f_{00}(p)dp, \ D_{01}(S) = \int_S f_{01}(p)dp, \ D_{10}(S) = \int_S f_{10}(p)dp, \ D_{11}(S) = \int_S f_{11}(p)dp.$$

Thus, $\text{gFDR}(S)$ is transformed into
$$\text{gFDR}(S) = \frac{\pi_{00}D_{00}(S) + \pi_{01}D_{01}(S) + \pi_{10}D_{10}(S)}{\pi_{00}D_{00}(S) + \pi_{01}D_{01}(S) + \pi_{10}D_{10}(S) + \pi_{11}D_{11}(S)}$$

$$= \frac{1}{1 + \{D_{11}(S)/(\gamma_{00}D_{00}(S) + \gamma_{01}D_{01}(S) + \gamma_{10}D_{10}(S))\}},$$
(A.2)

where $\gamma_{00} = \frac{\pi_{00}}{\pi_{11}}, \gamma_{01} = \frac{\pi_{01}}{\pi_{11}}, \gamma_{10} = \frac{\pi_{10}}{\pi_{11}}$. For any threshold $\zeta \in (0, 1]$, define the rejection region $S(\zeta)$ as

$$S(\zeta) = \left\{ p : \frac{\pi_{00}f_{00}(p) + \pi_{01}f_{01}(p) + \pi_{10}f_{10}(p)}{\pi_{00}f_{00}(p) + \pi_{01}f_{01}(p) + \pi_{10}f_{10}(p) + \pi_{11}f_{11}(p)} \le \zeta \right\}$$

$$= \left\{ p : \frac{1}{1 + \{f_{11}(p)/(\gamma_{00}f_{00}(p) + \gamma_{01}f_{01}(p) + \gamma_{10}f_{10}(p))\}} \le \zeta \right\}.$$
(A.3)

Here we prove that $\text{gFDR}(S(\zeta))$ is a non-decreasing function of $\zeta$. Suppose $\zeta_2 > \zeta_1$, considering two cases:

**Case 1**  $\nu(S(\zeta_2) - S(\zeta_1)) = 0$. We derive that $\text{gFDR}(S(\zeta_1)) = \text{gFDR}(S(\zeta_2))$.

**Case 2**  $\nu(S(\zeta_2) - S(\zeta_1)) > 0$. We can prove that $\text{gFDR}(S(\zeta))$ is a non-decreasing function of $\zeta$ if

$$\frac{D_{11}(S(\zeta_2) - S(\zeta_1))}{\gamma_{00}D_{00}(S(\zeta_2) - S(\zeta_1)) + \gamma_{01}D_{01}(S(\zeta_2) - S(\zeta_1)) + \gamma_{10}D_{10}(S(\zeta_2) - S(\zeta_1))}$$

$$< \frac{D_{11}(S(\zeta_1))}{\gamma_{00}D_{00}(S(\zeta_1)) + \gamma_{01}D_{01}(S(\zeta_1)) + \gamma_{10}D_{10}(S(\zeta_1))}$$
(A.4)

holds, the reason is as follows. Let
$$m_1 = \sup\left\{ \frac{f_{11}(p)}{\gamma_{00}f_{00}(p) + \gamma_{01}f_{01}(p) + \gamma_{10}f_{10}(p)} : p \in S(\zeta_2) - S(\zeta_1) \right\},$$

$$m_2 = \inf\left\{ \frac{f_{11}(p)}{\gamma_{00}f_{00}(p) + \gamma_{01}f_{01}(p) + \gamma_{10}f_{10}(p)} : p \in S(\zeta_1) \right\}.$$

By the definition of region $S(\zeta)$, we have $m_2 > m_1$ obviously. Therefore, we have

$$\frac{D_{11}(S(\zeta_1))}{\gamma_{00}D_{00}(S(\zeta_1)) + \gamma_{01}D_{01}(S(\zeta_1)) + \gamma_{10}D_{10}(S(\zeta_1))}$$

$$\ge m_2 \frac{\gamma_{00}D_{00}(S(\zeta_1)) + \gamma_{01}D_{01}(S(\zeta_1)) + \gamma_{10}D_{10}(S(\zeta_1))}{\gamma_{00}D_{00}(S(\zeta_1)) + \gamma_{01}D_{01}(S(\zeta_1)) + \gamma_{10}D_{10}(S(\zeta_1))}$$

$$> m_1 \frac{\gamma_{00}D_{00}(S(\zeta_2) - S(\zeta_1)) + \gamma_{01}D_{01}(S(\zeta_2) - S(\zeta_1)) + \gamma_{10}D_{10}(S(\zeta_2) - S(\zeta_1))}{\gamma_{00}D_{00}(S(\zeta_2) - S(\zeta_1)) + \gamma_{01}D_{01}(S(\zeta_2) - S(\zeta_1)) + \gamma_{10}D_{10}(S(\zeta_2) - S(\zeta_1))}$$
(A.5)

$$\ge \frac{D_{11}(S(\zeta_2) - S(\zeta_1))}{\gamma_{00}D_{00}(S(\zeta_2) - S(\zeta_1)) + \gamma_{01}D_{01}(S(\zeta_2) - S(\zeta_1)) + \gamma_{10}D_{10}(S(\zeta_2) - S(\zeta_1))}.$$

Furthermore, we decompose the region $S(\zeta_2)$ as follows

$$\frac{D_{11}(S(\zeta_2))}{\gamma_{00}D_{00}(S(\zeta_2))+\gamma_{01}D_{01}(S(\zeta_2))+\gamma_{10}D_{10}(S(\zeta_2))}$$

$$= \left\{D_{11}(S(\zeta_2)-S(\zeta_1))+D_{11}(S(\zeta_1))\right\} \Big/ \left\{\begin{array}{l} \gamma_{00}D_{00}(S(\zeta_2)-S(\zeta_1))+\gamma_{01}D_{01}(S(\zeta_2)-S(\zeta_1)) \\ +\gamma_{10}D_{10}(S(\zeta_2)-S(\zeta_1))+\gamma_{00}D_{00}(S(\zeta_1)) \\ +\gamma_{01}D_{01}(S(\zeta_1))+\gamma_{10}D_{10}(S(\zeta_1)) \end{array}\right\}. \tag{A.6}$$

Combined with (A.4), we obtain

$$\frac{D_{11}(S(\zeta_1))}{\gamma_{00}D_{00}(S(\zeta_1))+\gamma_{01}D_{01}(S(\zeta_1))+\gamma_{10}D_{10}(S(\zeta_1))} > \frac{D_{11}(S(\zeta_2))}{\gamma_{00}D_{00}(S(\zeta_2))+\gamma_{01}D_{01}(S(\zeta_2))+\gamma_{10}D_{10}(S(\zeta_2))}. \tag{A.7}$$

Moreover, by the definition of $\mathrm{gFDR}(S)$, it holds that

$$\mathrm{gFDR}(S(\zeta_1)) < \mathrm{gFDR}(S(\zeta_2)).$$

Under the Assumption (ii) in Theorem 1, for a given $\alpha \in (0,1)$, there exists a threshold $\zeta^\star > 0$, s.t. $\mathrm{gFDR}(S(\zeta^\star)) = \alpha$. For the ease of presentation, we denote $S(\zeta^\star)$ as $S^\star$. In the following, we will prove that $S^\star$ is the optimal rejection region.

Considering any set $T$ that satisfies $D_{11}(T) > D_{11}(S^\star)$. Let $R_T = T - S^\star$ and $R_S = S^\star - T$. We can derive that

$$\begin{aligned} D_{11}(T) &= D_{11}(T \cap S^\star) + D_{11}(R_T), \\ D_{11}(S) &= D_{11}(T \cap S^\star) + D_{11}(R_S). \end{aligned} \tag{A.8}$$

Then, we have $D_{11}(R_T) > D_{11}(R_S)$. By the definition of $S^\star$, we have

$$\inf\left\{\frac{\gamma_{00}f_{00}(p)+\gamma_{01}f_{01}(p)+\gamma_{10}f_{10}(p)}{f_{11}(p)} : p \in R_T\right\} > \sup\left\{\frac{\gamma_{00}f_{00}(p)+\gamma_{01}f_{01}(p)+\gamma_{10}f_{10}(p)}{f_{11}(p)} : p \in R_S\right\}. \tag{A.9}$$

Therefore,

$$\frac{\gamma_{00}D_{00}(R_T)+\gamma_{01}D_{01}(R_T)+\gamma_{10}D_{10}(R_T)}{D_{11}(R_T)} > \frac{\gamma_{00}D_{00}(R_S))+\gamma_{01}D_{01}(R_S)+\gamma_{10}D_{10}(R_S)}{D_{11}(R_S)}. \tag{A.10}$$

In a similar way, we can derive that

$$\frac{\gamma_{00}D_{00}(R_T)+\gamma_{01}D_{01}(R_T)+\gamma_{10}D_{10}(R_T)}{D_{11}(R_T)} > \frac{\gamma_{00}D_{00}(T\cap S^\star))+\gamma_{01}D_{01}((T\cap S^\star)+\gamma_{10}D_{10}((T\cap S^\star)}{D_{11}((T\cap S^\star)}. \tag{A.11}$$

Finally, we have

$$\frac{\gamma_{00}D_{00}(T)+\gamma_{01}D_{01}(T)+\gamma_{10}D_{10}(T)}{D_{11}(T)}$$

$$= \frac{\gamma_{00}D_{00}(T\cap S^\star)+\gamma_{01}D_{01}(T\cap S^\star)+\gamma_{10}D_{10}(T\cap S^\star)+\gamma_{00}D_{00}(R_T)+\gamma_{01}D_{01}(R_T)+\gamma_{10}D_{10}(R_T)}{D_{11}(T\cap S^\star)+D_{11}(R_T)}$$

$$> \frac{\gamma_{00}D_{00}(T\cap S^\star)+\gamma_{01}D_{01}(T\cap S^\star)+\gamma_{10}D_{10}(T\cap S^\star)+\gamma_{00}D_{00}(R_S)+\gamma_{01}D_{01}(R_S)+\gamma_{10}D_{10}(R_S)}{D_{11}(T\cap S^\star)+D_{11}(R_S)}$$

$$= \frac{\gamma_{00}D_{00}(S^\star)+\gamma_{01}D_{01}(S^\star)+\gamma_{10}D_{10}(S^\star)}{D_{11}(S^\star)}. \tag{A.12}$$

The second inequality holds because $D_{11}(R_T) > D_{11}(R_S)$, implying $\mathrm{gFDR}(T) > \mathrm{gFDR}(S(\zeta^\star)) = \alpha$. Therefore, we can conclude that the rejection region $S(\zeta^\star)$ is optimal.

### E.2 Proof of Theorem 2

#### E.2.1 The consistent estimator of local FDR

To justify Assumption (i) for the corresponding local FDR estimator in Theorem 2, we first prove the consistency of local FDR estimator under $L_\infty$ norm in Proposition 1.

**Proposition 1.** *Assume that the smoothing parameter $b$ satisfies*

$$\lim_{J \to \infty} b = 0 \text{ and } \lim_{J \to \infty} Jb^2 = +\infty.$$

*Then, we have*

$$\sup_{p_j \in [0,1]^2} \left| \widehat{\text{fdr}}(p_j) - \text{fdr}(p_j) \right| \xrightarrow{P} 0 \text{ as } n, J \to \infty.$$

Let $g$ be a probability density on $[0,1]$, and $\hat{g}$ be the beta kernel estimator:

$$\hat{g}(p^{(i)}) = J^{-1} \sum_{j=1}^{J} K^{\star}_{p^{(i)},b}(p_{ij}), \quad i = 1, 2.$$

To prove the consistency of the beta kernel estimator $\hat{g}$, i.e

$$\sup_{p^{(i)} \in [0,1]} \left| \hat{g}(p^{(i)}) - g(p^{(i)}) \right| \xrightarrow{P} 0 \text{ as } J \to \infty, \tag{A.13}$$

we first need to establish the uniform convergence of its bias on the interval $[0,1]$.
**Lemma 1.** *Let $g$ be the probability density on $[0,1]$, and $\hat{g}$ be the beta kernel estimator. We have*

$$\sup_{p^{(i)} \in [0,1]} \left| \text{E} \left\{ \hat{g}(p^{(i)}) \right\} - g(p^{(i)}) \right| \to 0 \text{ as } b \to 0, \quad i = 1, 2.$$

*Proof of Lemma 1.* Without loss of generality, we replace $p^{(i)}, i = 1, 2$ with $p$ for simplifying the proof steps, and discuss three cases in the following.

**Case 1** $p \in (2b, 1 - 2b)$  Denote $\mu_1$ and $\sigma_1^2$ are mean and variance of P, a variable following Beta$(p/b, (1-p)/b)$. According to Johnson et al. [26], there exists a constant $C$ such that

$$\mu_1 = p, \tag{A.14}$$

$$\sigma_1^2 = bp(1-p) + R_2(p), \tag{A.15}$$

where $R_2(p) \leq Cb^2$. Because $f$ is a probability density on $[0,1]$, for $\varepsilon > 0$, there exists a $\delta > 0$ such that

$$|g(t) - g(p)| < \varepsilon \quad \text{for } |p - t| < \delta \tag{A.16}$$

for all $p \in (2b, 1 - 2b)$; According to (A.14), we have

$$|\mu_1 - p| < \delta/2 \text{ for all } p \in (2b, 1 - 2b). \tag{A.17}$$

Therefore, we can derive that

$$
\begin{aligned}
|E\{\hat{g}(p)\} - g(p)| &= \left| \int_{2b}^{1-2b} \{g(t) - g(p)\} K\left(t, \frac{p}{b}, \frac{1-p}{b}\right) dt \right| \\
&\leq \int_{|t-\mu_1|<\delta/2} |g(t) - g(p)| K\left(t, \frac{p}{b}, \frac{1-p}{b}\right) dt \\
&\quad + \int_{|t-\mu_1|>\delta/2} |g(t) - g(p)| K\left(t, \frac{p}{b}, \frac{1-p}{b}\right) dt \\
&\leq \int_{|t-\mu_1|<\delta/2} |g(t) - g(p)| K\left(t, \frac{p}{b}, \frac{1-p}{b}\right) dt \\
&\quad + 2 \sup_{t \in (2b, 1-2b)} |g(t)| \int_{|t-\mu_1|>\delta/2} K\left(t, \frac{p}{b}, \frac{1-p}{b}\right) dt \\
&\equiv \mathcal{M}_1 + \mathcal{M}_2.
\end{aligned}
$$

According (A.16) and (A.17), we obtain

$$\mathcal{M}_1 \leq \varepsilon. \tag{A.18}$$

Combining the Chebyshev's inequality and (A.15), and there also exists $b_\varepsilon$ such that

$$\mathcal{M}_2 \leq \left\{ 8 \sup_{t\in(2b,1-2b)} |g(t)|\sigma_1^2 \right\} /\delta^2 \leq \left\{ 2 \sup_{t\in(2b,1-2b)} |g(t)| \left(b + 4Cb^2\right) \right\} /\delta^2 \leq \varepsilon \text{ for all } b \leq b_\varepsilon.$$

(A.19)

Thus, from (A.18) and (A.19), we conclude that

$$\sup_{p\in(2b,1-2b)} |\mathrm{E}\{\hat{g}(p)\} - g(p)| < 2\varepsilon \quad \text{for all } b \leq b_\varepsilon.$$

**Case 2** $p \in [0, 2b]$  Based on the notations of Case 1, we have

$$\mu_2 = p + \xi(p,b),$$

(A.20)

$$\sigma_2^2 = R_2(p),$$

(A.21)

where $\mu_2$ and $\sigma_2^2$ are mean and variance of of P, a variable following Beta $(\rho(p,b), (1-p)/b)$, $\xi(p,b) = (1-p)\{\rho(p,b) - p/b\}/\{1 + b\rho(p,b) - p\}$, and $R_2(p) \leq Cb^2$. For $\varepsilon > 0$, there exists a $\delta > 0$ such that

$$|g(t) - g(p)| < \varepsilon \quad \text{for } |p - t| < \delta$$

(A.22)

for all $p \in [0, 2b]$; According to (A.20), since $\xi(p,b)$ is a bounded function for $p \in [0, 2b]$, there also exists $b_\delta$ such that

$$|\mu_2 - p| < \delta/2 \text{ for } b \leq b_\delta \quad \text{for all } p \in [0, 2b].$$

(A.23)

Therefore, we can derive that

$$
\begin{aligned}
|\mathrm{E}\{\hat{g}(p)\} - g(p)| &= \left| \int_0^{2b} \{g(t) - g(p)\} K\left(t, \rho(p,b), \frac{1-p}{b}\right) dt \right| \\
&\leq \int_{|t-\mu_2|<\delta/2} |g(t) - g(p)| K\left(t, \rho(p,b), \frac{1-p}{b}\right) dt \\
&\quad + \int_{|t-\mu_2|>\delta/2} |g(t) - g(p)| K\left(t, \rho(p,b), \frac{1-p}{b}\right) dt \\
&\leq \int_{|t-\mu_2|<\delta/2} |g(t) - g(p)| K\left(t, \rho(p,b), \frac{1-p}{b}\right) dt \\
&\quad + 2 \sup_{t\in[0,2b]} |g(t)| \int_{|t-\mu_2|>\delta/2} K\left(t, \rho(p,b), \frac{1-p}{b}\right) dt \\
&\equiv \mathcal{M}_1 + \mathcal{M}_2.
\end{aligned}
$$

According to (A.22) and (A.23), there exists $b_\varepsilon^{(1)}$ such that

$$\mathcal{M}_1 \leq \varepsilon \text{ for all } b \leq b_\varepsilon^{(1)}.$$

(A.24)

Combining the Chebyshev's inequality and (A.21), and there also exists $b_\varepsilon^{(2)}$ such that

$$\mathcal{M}_2 \leq \left\{ 8 \sup_{t\in[0,2b]} |g(t)|\sigma_2^2 \right\} /\delta^2 \leq \left\{ 8 \sup_{t\in[0,2b]} |g(t)|Cb^2 \right\} /\delta^2 \leq \varepsilon \text{ for all } b \leq b_\varepsilon^{(2)}.$$

(A.25)

Thus, from (A.24) and (A.25), we conclude that

$$\sup_{p\in[0,2b]} |\mathrm{E}\{\hat{g}(p)\} - g(p)| < 2\varepsilon \quad \text{for all } b \leq \min\left(b_\varepsilon^{(1)}, b_\varepsilon^{(2)}\right).$$

**Case 3** $p \in [1 - 2b, 1]$  Case 3 can be proven a similar procedure. We note that

$$\mu_3 = p - b \cdot \xi(1 - p, b),$$

(A.26)

$$\sigma_3^2 = R_2(p),$$

(A.27)

where $\mu_3$ and $\sigma_3^2$ are mean and variance of P, a variable following Beta$(p/b, \rho(1-p, b))$, and $R_2(p) \leq Cb^2$.

This completes the proof of Lemma 1. $\qquad\square$

*Proof of Proposition 1.* To prove the consistency of the beta kernel estimator, we use the inequality:

$$\sup_{p\in[0,1]} |\hat{g}(p) - g(p)| \le \sup_{p\in[0,1]} |\hat{g}(p) - \mathrm{E}\{\hat{g}(p)\}| + \sup_{p\in[0,1]} |\mathrm{E}\{\hat{g}(p)\} - g(p)|. \tag{A.28}$$

From Lemma 1, the second term converges to zero. In the following, we prove that

$$\sup_{p\in[0,1]} |\hat{g}(p) - \mathrm{E}\{\hat{g}(p)\}| \xrightarrow{P} 0 \text{ as } J \to \infty. \tag{A.29}$$

We also consider three cases:

**Case 1** $p \in (2b, 1-2b)$   The beta kernel estimator $\hat{g}(p)$ is expressed as

$$\hat{g}(p) = \int_{2b}^{1-2b} K\left(t, \frac{p}{b}, \frac{1-p}{b}\right) dF_n(t), \tag{A.30}$$

where $F_n$ is the empirical distribution. The expectation of the beta kernel estimator is

$$\mathrm{E}\{\hat{g}(p)\} = \int_{2b}^{1-2b} K\left(t, \frac{p}{b}, \frac{1-p}{b}\right) dF(t). \tag{A.31}$$

Thus, for $p \in (2b, 1-2b)$, we can derive that

$$|\hat{g}(p) - \mathrm{E}\{\hat{g}(p)\}| = \left| \int_{2b}^{1-2b} K\left(t, \frac{p}{b}, \frac{1-p}{b}\right) d\{F_n(t) - F(t)\} \right| \tag{A.32}$$

$$\le \sup_{t\in(b,1-2b)} |F_n(t) - F(t)| \int_{2b}^{1-2b} \left| dK\left(t, \frac{p}{b}, \frac{1-p}{b}\right) \right|.$$

Note that the integral in (A.32) is bounded above by

$$\frac{1-b}{b} \int_{2b}^{1-2b} \left| K\left(t, \frac{p}{b}-1, \frac{1-p}{b}\right) - K\left(t, \frac{p}{b}, \frac{1-p}{b}-1\right) \right| dt \le 2\frac{1-b}{b}. \tag{A.33}$$

Therefore,

$$|\hat{g}(p) - \mathrm{E}\{\hat{g}(p)\}| \le 2\frac{1-b}{b} \sup_{t\in(2b,1-2b)} |F_n(t) - F(t)|. \tag{A.34}$$

From Dvoretzky et al. [16], we can obtain

$$\mathrm{P}\left[\sup_{p\in(2b,1-2b)} |\hat{g}(p) - \mathrm{E}\{\hat{g}(p)\}| \ge \varepsilon\right] \le \mathrm{P}\left\{\sup_{t\in(2b,1-2b)} |F_n(t) - F(t)| \ge \frac{\varepsilon}{2} \cdot \frac{b}{1-b}\right\} \tag{A.35}$$

$$\le 2\exp\left\{-J\frac{\varepsilon^2}{2}\frac{b^2}{(1-b)^2}\right\}.$$

By utilizing the Borel-Cantelli Lemma, it is shown that under the beta kernel estimator is consistent.

**Case 2** $p \in [0, 2b]$   Case 2 can be proven a similar procedure of Case 1. Note that, for all $p \in [0, 2b]$,

$$|\hat{g}(p) - \mathrm{E}\{\hat{g}(p)\}| = \left| \int_0^{2b} K\left(t, \rho(p,b), \frac{1-p}{b}\right) d\{F_n(t) - F(t)\} \right| \tag{A.36}$$

$$\le \sup_{t\in[0,2b]} |F_n(t) - F(t)| \int_0^{2b} \left| dK\left(t, \rho(p,b), \frac{1-p}{b}\right) \right|.$$

Since $\rho(p,b)$ is monotonic increasing in $[0, 2b]$, $\rho(0,b) = 1$, $\rho(2b,b) = 2$. For $p \in (0, 2b]$, the integral in (A.36) is bounded above by $2\frac{1+b}{b}$. For $p = 0$, it is bounded above by

$$\left(\rho(p,b) + \frac{1-p}{b} - 1\right) \int_0^{2b} \left| K\left(t, \rho(p,b), \frac{1-p}{b}-1\right) \right| dt = \frac{1+b}{b}. \tag{A.37}$$

Thus, we can obtain

$$\mathrm{P}\left[\sup_{p\in[0,2b]}|\hat{g}(p)-\mathrm{E}\{\hat{g}(p)\}|\geq\varepsilon\right]\leq\mathrm{P}\left\{\sup_{t\in[0,2b]}|F_n(t)-F(t)|\geq\frac{\varepsilon}{2}\cdot\frac{b}{1+b}\right\}$$
$$\leq 2\exp\left\{-J\frac{\varepsilon^2}{2}\frac{b^2}{(1+b)^2}\right\},\qquad\text{(A.38)}$$

which concludes the proof of the consistency of beta kernel estimator in Case 2.

**Case 3** $p\in[1-2b,1]$   Case 3 can be proven a similar procedure of Case 1. Note that for all $p\in[1-2b,1]$,

$$|\hat{g}(p)-\mathrm{E}\{\hat{g}(p)\}|=\left|\int_{1-2b}^1 K\left(t,\frac{p}{b},\rho(1-p,b)\right)d\{F_n(t)-F(t)\}\right|$$
$$\leq\sup_{t\in[1-2b,1]}|F_n(t)-F(t)|\int_{1-2b}^1\left|dK\left(t,\frac{p}{b},\rho(1-p,b)\right)\right|.\qquad\text{(A.39)}$$

For $p\in[1-2b,1)$, the integral in (A.39) is bounded above by $2\frac{1+b}{b}$. And for $p=1$, it is bounded above by

$$(\rho(1-p,b)+\frac{p}{b}-1)\int_{1-2b}^1\left|K\left(t,\frac{p}{b}-1,\rho(1-p,b)\right)\right|dt=\frac{1+b}{b}.\qquad\text{(A.40)}$$

Thus for all $p\in[1-2b,1]$,

$$|\hat{g}(p)-E\{\hat{g}(p)\}|\leq 2\frac{1+b}{b}\sup_{t\in[1-2b,1]}|F_n(t)-F(t)|.\qquad\text{(A.41)}$$

Similarly, we obtain

$$\mathrm{P}\left[\sup_{p\in[1-2b,1]}|\hat{g}(p)-\mathrm{E}\{\hat{g}(p)\}|\geq\varepsilon\right]\leq\mathrm{P}\left\{\sup_{t\in[1-2b,1]}|F_n(t)-F(t)|\geq\frac{\varepsilon}{2}\cdot\frac{b}{1+b}\right\}$$
$$\leq 2\exp\left\{-J\frac{\varepsilon^2}{2}\frac{b^2}{(1+b)^2}\right\},\qquad\text{(A.42)}$$

which concludes the proof of the consistency of beta kernel estimator in Case 3.

From Dai et al. [12], for a fixed $J$ and $\lambda$, the biases of $\hat{\pi}_{00}$, $\hat{\pi}_{10}$, and $\hat{\pi}_{01}$ go to zero as $n\to\infty$:

$$\lim_{n\to\infty}\hat{\pi}_{00}=\pi_{00},\quad\lim_{n\to\infty}\hat{\pi}_{01}=\pi_{01},\quad\lim_{n\to\infty}\hat{\pi}_{10}=\pi_{10}.\qquad\text{(A.43)}$$

And we can derive

$$\hat{f}(p)=\hat{f}(p^{(1)},p^{(2)})=\hat{g}(p^{(1)})\cdot\hat{g}(p^{(2)}).\qquad\text{(A.44)}$$

By combining equations (A.13), (A.43), and (A.44), according to continuous mapping theorem [14], we have

$$\sup_{p\in[0,1]}\left|\widehat{\mathrm{fdr}}(p)-\mathrm{fdr}(p)\right|\xrightarrow{P}0\text{ as }n,J\to\infty.\qquad\text{(A.45)}$$

$$\square$$

According to equation (A.45), we can verify the rationality of Assumption (i) in Theorem 2:

$$\frac{1}{J}\sum_{j=1}^J\left|\widehat{\mathrm{fdr}}(p_j)-\mathrm{fdr}(p_j)\right|\xrightarrow{P}0\text{ as }n,J\to\infty.\qquad\text{(A.46)}$$

### E.2.2 Proof of Theorem 2

To begin with, we introduce some notations. For $\zeta \in (0,1]$, denote

$$\widehat{G}_J^{00}(\zeta) = \frac{1}{J_{00}} \sum_{j \in \Delta_{00}} \mathbb{I}\left(p_j \in \widehat{S}(\zeta)\right), \quad \widehat{G}_J^{01}(\zeta) = \frac{1}{J_{01}} \sum_{j \in \Delta_{01}} \mathbb{I}\left(p_j \in \widehat{S}(\zeta)\right),$$

$$\widehat{G}_J^{10}(\zeta) = \frac{1}{J_{10}} \sum_{j \in \Delta_{10}} \mathbb{I}\left(p_j \in \widehat{S}(\zeta)\right), \quad \widehat{G}_J^{11}(\zeta) = \frac{1}{J_{11}} \sum_{j \in \Delta_{11}} \mathbb{I}\left(p_j \in \widehat{S}\zeta\right),$$

$$G_J^{00}(\zeta) = \frac{1}{J_{00}} \sum_{j \in \Delta_{00}} \mathbb{P}\left(p_j \in \widehat{S}(\zeta)\right), \quad G_J^{01}(\zeta) = \frac{1}{J_{01}} \sum_{j \in \Delta_{01}} \mathbb{P}\left(p_j \in \widehat{S}(\zeta)\right),$$

$$G_J^{10}(\zeta) = \frac{1}{J_{10}} \sum_{j \in \Delta_{10}} \mathbb{P}\left(p_j \in \widehat{S}(\zeta)\right),$$

$$\widehat{V}_J^{00}(\zeta) = \frac{1}{J_{00}} \sum_{j \in \Delta_{00}} \mathbb{I}\left(p_j \in \tilde{S}_{00}(\zeta)\right), \quad \widehat{V}_J^{01}(\zeta) = \frac{1}{J_{01} + J_{00}} \sum_{j \in \Delta_{01} \cup \Delta_{00}} \mathbb{I}\left(p_j \in \tilde{S}_{01}(\zeta)\right),$$

$$\widehat{V}_J^{10}(\zeta) = \frac{1}{J_{10} + J_{00}} \sum_{j \in \Delta_{10} \cup \Delta_{00}} \mathbb{I}\left(p_j \in \tilde{S}_{10}(\zeta)\right),$$

where $J_{00} = |\Delta_{00}|$, $J_{01} = |\Delta_{01}|$, $J_{10} = |\Delta_{10}|$, $J_{11} = |\Delta_{11}|$. Denote $r_J^{00} = J_{00}/J_{11}, r_J^{01} = J_{01}/J_{11}, r_J^{10} = J_{10}/J_{11}, v_J = \dfrac{J_{11}}{J_{00} + J_{01} + J_{10}}$. And

$$\overline{K}_J^0(\zeta) = v_J\{r_J^{00}G_J^{00}(\zeta) + r_J^{01}G_J^{01}(\zeta) + r_J^{10}G_J^{10}(\zeta)\},$$

$$K_J^0(\zeta) = v_J\{r_J^{00}\widehat{G}_J^{00}(\zeta) + r_J^{01}\widehat{G}_J^{01}(\zeta) + r_J^{10}\widehat{G}_J^{10}(\zeta)\},$$

$$\widehat{K}_J^0(\zeta) = v_J\{(r_J^{01} + r_J^{00})\widehat{V}_J^{01}(\zeta) + (r_J^{10} + r_J^{00})\widehat{V}_J^{10}(\zeta) - r_J^{00}\widehat{V}_J^{00}(\zeta)\},$$

$$\text{FDP}_J(\zeta) = \frac{r_J^{00}\widehat{G}_J^{00}(\zeta) + r_J^{01}\widehat{G}_J^{01}(\zeta) + r_J^{10}\widehat{G}_J^{10}(\zeta)}{r_J^{00}\widehat{G}_J^{00}(\zeta) + r_J^{01}\widehat{G}_J^{01}(\zeta) + r_J^{10}\widehat{G}_J^{10}(\zeta) + \widehat{G}_J^{11}(\zeta)},$$

$$\text{FDP}_J^\dagger(\zeta) = \frac{\widehat{K}_J^0(\zeta)/v_J}{r_J^{00}\widehat{G}_J^{00}(\zeta) + r_J^{01}\widehat{G}_J^{01}(\zeta) + r_J^{10}\widehat{G}_J^{10}(\zeta) + \widehat{G}_J^{11}(\zeta)},$$

$$\overline{\text{FDP}}_J(\zeta) = \frac{\overline{K}_J^0(\zeta)/v_J}{r_J^{00}G_J^{00}(\zeta) + r_J^{01}G_J^{01}(\zeta) + r_J^{10}G_J^{10}(\zeta) + \widehat{G}_J^{11}(\zeta)}.$$

Before proceeding with the proof of Theorem 2, we prove Lemma 2 first.

**Lemma 2.** *Under Assumption (i)-(ii) in Theorem 2, if $J_{00} \to \infty$, $J_{01} \to \infty$, $J_{10} \to \infty$ as $J \to \infty$, and $n \to \infty$, we have in probability,*

$$\sup_{\zeta \in (0,1]} \left|\widehat{G}_J^{00}(\zeta) - G_J^{00}(\zeta)\right| \longrightarrow 0, \quad \sup_{\zeta \in (0,1]} \left|\widehat{G}_J^{01}(\zeta) - G_J^{01}(\zeta)\right| \longrightarrow 0,$$

$$\sup_{\zeta \in (0,1]} \left|\widehat{G}_J^{10}(\zeta) - G_J^{10}(\zeta)\right| \longrightarrow 0, \quad \sup_{\zeta \in (0,1]} \left|\widehat{K}_J^0(\zeta) - \overline{K}_J^0(\zeta)\right| \longrightarrow 0.$$

*Proof of Lemma 2.* We consider three cases under composite null hypothesis.

**Case 1** Under $H_{00}$: We can derive that

$$\widehat{G}_J^{00}(\zeta) = \frac{1}{J_{00}} \sum_{j \in \Delta_{00}} \left\{\mathbb{I}\left(p_j \in \widehat{S}(\zeta)\right) - \mathbb{I}\left(p_j \in S(\zeta)\right) + \mathbb{I}\left(p_j \in S(\zeta)\right)\right\},$$

$$G_J^{00}(\zeta) = \frac{1}{J_{00}} \sum_{j \in \Delta_{00}} \left\{\mathbb{P}\left(p_j \in \widehat{S}(\zeta)\right) - \mathbb{P}\left(p_j \in S(\zeta)\right) + \mathbb{P}\left(p_j \in S(\zeta)\right)\right\}.$$

$$(A.47)$$

Thus, we have

$$
\begin{aligned}
\sup_{\zeta\in(0,1]}\left|\widehat{G}_J^{00}(\zeta)-G_J^{00}(\zeta)\right| &\leq \sup_{\zeta\in(0,1]}\frac{1}{J_{00}}\sum_{j\in\Delta_{00}}\left|\mathbb{I}\left(p_j\in\widehat{S}(\zeta)\right)-\mathbb{I}\left(p_j\in S(\zeta)\right)\right| \\
&\quad+\sup_{\zeta\in(0,1]}\frac{1}{J_{00}}\sum_{j\in\Delta_{00}}\left|\mathbb{P}\left(p_j\in\widehat{S}(\zeta)\right)-\mathbb{P}\left(p_j\in S(\zeta)\right)\right| \\
&\quad+\sup_{\zeta\in(0,1]}\frac{1}{J_{00}}\sum_{j\in\Delta_{00}}\left|\mathbb{I}\left(p_j\in S(\zeta)\right)-\mathbb{P}\left(p_j\in S(\zeta)\right)\right|.
\end{aligned} \tag{A.48}
$$

To deal with the first term, we have

$$
\begin{aligned}
&\frac{1}{J_{00}}\sum_{j\in\Delta_{00}}\left|\mathbb{I}\left(p_j\in\widehat{S}(\zeta)\right)-\mathbb{I}\left(p_j\in S(\zeta)\right)\right| \\
&=\frac{1}{J_{00}}\sum_{j\in\Delta_{00}}\left|\mathbb{I}\left\{\widehat{\mathrm{fdr}}\left(p_j\right)\leq\zeta\right\}-\mathbb{I}\left\{\mathrm{fdr}\left(p_j\right)\leq\zeta\right\}\right| \\
&=\frac{1}{J_{00}}\sum_{j\in\Delta_{00}}\left[\mathbb{I}\left\{\widehat{\mathrm{fdr}}\left(p_j\right)\leq\zeta,\mathrm{fdr}\left(p_j\right)>\zeta\right\}+\mathbb{I}\left\{\mathrm{fdr}\left(p_j\right)\leq\zeta,\widehat{\mathrm{fdr}}\left(p_j\right)>\zeta\right\}\right] \\
&=\frac{1}{J_{00}}\sum_{j\in\Delta_{00}}\left[\mathbb{I}\left\{\widehat{\mathrm{fdr}}\left(p_j\right)\leq\zeta,\zeta+\epsilon\geq\mathrm{fdr}\left(p_j\right)>\zeta\right\}+\mathbb{I}\left\{\zeta-\epsilon<\mathrm{fdr}\left(p_j\right)\leq\zeta,\widehat{\mathrm{fdr}}\left(p_j\right)>\zeta\right\}\right] \\
&\quad+\frac{1}{J_{00}}\sum_{j\in\Delta_{00}}\left[\mathbb{I}\left\{\widehat{\mathrm{fdr}}\left(p_j\right)\leq\zeta,\mathrm{fdr}\left(p_j\right)>\zeta+\epsilon\right\}+\mathbb{I}\left\{\mathrm{fdr}\left(p_j\right)\leq\zeta-\epsilon,\widehat{\mathrm{fdr}}\left(p_j\right)>\zeta\right\}\right] \\
&\leq\frac{1}{J_{00}}\sum_{j\in\Delta_{00}}\mathbb{I}\left\{\zeta-\epsilon<\mathrm{fdr}\left(p_j\right)\leq\zeta+\epsilon\right\}+\frac{1}{J_{00}\epsilon}\sum_{j\in\Delta_{00}}\left|\widehat{\mathrm{fdr}}\left(p_j\right)-\mathrm{fdr}\left(p_j\right)\right|.
\end{aligned} \tag{A.49}
$$

Combine with the Glivenko-Cantelli theorem and Assumption (i), we can derive

$$
\begin{aligned}
Q:=&\sup_{\zeta\in(0,1]}\frac{1}{J_{00}}\sum_{j\in\Delta_{00}}\left|\mathbb{I}\left\{\widehat{\mathrm{fdr}}\left(p_j\right)\leq\zeta\right\}-\mathbb{I}\left\{\mathrm{fdr}\left(p_j\right)\leq\zeta\right\}\right| \\
\leq&\sup_{\zeta\in(0,1]}\frac{1}{J_{00}}\sum_{j\in\Delta_{00}}\mathbb{I}\left\{\zeta-\epsilon<\mathrm{fdr}\left(p_j\right)\leq\zeta+\epsilon\right\}+\frac{1}{J_{00}\epsilon}\sum_{j\in\Delta_{00}}\left|\widehat{\mathrm{fdr}}\left(p_j\right)-\mathrm{fdr}\left(p_j\right)\right| \\
\leq&\sup_{\zeta\in(0,1]}\frac{1}{J_{00}}\sum_{j\in\Delta_{00}}\left|\mathbb{P}(\zeta-\epsilon<\mathrm{fdr}\left(p_j\right)\leq\zeta+\epsilon)\right| \\
&+2\sup_{\zeta\in(0,1]}\left|\frac{1}{J_{00}}\sum_{j\in\Delta_{00}}\mathbb{I}\left\{\mathrm{fdr}\left(p_j\right)\leq\zeta\right\}-\frac{1}{J_{00}}\sum_{j\in\Delta_{00}}\mathbb{P}\left\{\mathrm{fdr}\left(p_j\right)\leq\zeta\right\}\right| \\
&+\frac{1}{J_{00}\epsilon}\sum_{j\in\Delta_{00}}\left|\widehat{\mathrm{fdr}}\left(p_j\right)-\mathrm{fdr}\left(p_j\right)\right| \\
\leq&\sup_{\zeta\in(0,1]}\frac{1}{J_{00}}\sum_{j\in\Delta_{00}}\left|\mathbb{P}(\zeta-\epsilon<\mathrm{fdr}\left(p_j\right)\leq\zeta+\epsilon)\right|+o_p(1).
\end{aligned}
$$

Since $\epsilon$ can be arbitrarily small, $\sup_{\zeta\in(0,1]}\frac{1}{J_{00}}\sum_{j\in\Delta_{00}}|\mathbb{P}(\zeta-\epsilon<\mathrm{fdr}\left(p_j\right)\leq\zeta+\epsilon)|$ can be small due to Assumption (ii). Consequently, we have $Q=o_p(1)$ and thus the first term holds.

Before addressing the second term, we obtain that

$$
\begin{aligned}
&\mathbb{P}\left(\widehat{\mathrm{fdr}}\left(p_j\right)\leq\zeta\right) \\
\leq&\mathbb{P}\left(\widehat{\mathrm{fdr}}\left(p_j\right)\leq\zeta,\mathrm{fdr}\left(p_j\right)\leq\zeta+\epsilon\right)+\mathbb{P}\left(\widehat{\mathrm{fdr}}\left(p_j\right)\leq\zeta,\mathrm{fdr}\left(p_j\right)>\zeta+\epsilon\right) \\
\leq&\mathbb{P}\left(\mathrm{fdr}\left(p_j\right)\leq\zeta+\epsilon\right)+\mathbb{P}\left(\left|\widehat{\mathrm{fdr}}\left(p_j\right)-\mathrm{fdr}\left(p_j\right)\right|>\epsilon\right).
\end{aligned} \tag{A.50}
$$

Combine with Assumption (i), we can derive that $\mathbb{P}\left(\left|\widehat{\mathrm{fdr}}\left(p_j\right) - \mathrm{fdr}\left(p_j\right)\right| > \epsilon\right) \to 0$.

Then, we have

$$
\begin{aligned}
&\sup_{\zeta \in (0,1]} \sum_{j \in \Delta_{00}} \left|\mathbb{P}\left(p_j \in \widehat{S}(\zeta)\right) - \mathbb{P}\left(p_j \in S(\zeta)\right)\right| \\
&= \sup_{\zeta \in (0,1]} \frac{1}{J_{00}} \sum_{j \in \Delta_{00}} \left|\mathbb{P}\left(\widehat{\mathrm{fdr}}\left(p_j\right) \leq \zeta\right) - \mathbb{P}\left(\mathrm{fdr}\left(p_j\right) \leq \zeta\right)\right| \\
&\leq \sup_{\zeta \in (0,1]} \frac{1}{J_{00}} \sum_{j \in \Delta_{00}} \left|\mathbb{P}\left(\mathrm{fdr}\left(p_j\right) \leq \zeta + \epsilon\right) - \mathbb{P}\left(\mathrm{fdr}\left(p_j\right) \leq \zeta\right)\right| \\
&= \sup_{\zeta \in (0,1]} \frac{1}{J_{00}} \sum_{j \in \Delta_{00}} \left|\mathbb{P}\left(\zeta < \mathrm{fdr}\left(p_j\right) \leq \zeta + \epsilon\right)\right|.
\end{aligned}
\tag{A.51}
$$

As $\epsilon$ can be arbitrarily small, $\sup_{\zeta \in (0,1]} \frac{1}{J_{00}} \sum_{j \in \Delta_{00}} \left|\mathbb{P}\left(\zeta < \mathrm{fdr}\left(p_j\right) \leq \zeta + \epsilon\right)\right| \to 0$.

The third term can be proved using the Glivenko-Cantelli theorem. Thus, we have shown the proof of the first claim in Lemma 2.

**Cases 2-3**  Under $H_{01}$ and $H_{10}$: Following the similar procedure in Case 1, we can conclude the proof of the second and third claims in Lemma 2.

**Case 4**  According to the symmetric property of p-values $p_j$ for $j \in \Delta_{00} \cup \Delta_{01} \cup \Delta_{10}$, we follow the similar steps in Case 1, thus, we have

$$
\sup_{\zeta \in (0,1]} \left|\widehat{K}_J^0(\zeta) - \overline{K}_J^0(\zeta)\right| \longrightarrow 0.
\tag{A.52}
$$

This concludes the proof of the fourth claim.

$\square$

*Proof of Theorem 2.*  For any $\epsilon \in (0, \alpha)$, suppose there exists $\zeta_{\alpha-\epsilon} > 0$, then

$$
\mathbb{P}\left(\mathrm{FDP}\left(\zeta_{\alpha-\epsilon}\right) \leq \alpha - \epsilon\right) \to 1.
$$

By Lemma 2, for any constant $c > 0$, we have

$$
\sup_{0 < \zeta \leq c} \left|\mathrm{FDP}_J^\dagger(\zeta) - \mathrm{FDP}_J(\zeta)\right| \xrightarrow{p} 0.
\tag{A.53}
$$

By the definition of $\zeta^\star$, i.e., $\zeta^\star = \sup\left\{\zeta \in (0,1] : \mathrm{FDP}_J^\dagger(\zeta) \leq \alpha\right\}$, we have

$$
\begin{aligned}
\mathbb{P}\left(\zeta^\star \geq \zeta_{\alpha-\epsilon}\right) &\geq \mathbb{P}\left(\mathrm{FDP}_J^\dagger\left(\zeta_{\alpha-\epsilon}\right) \leq \alpha\right) \\
&\geq \mathbb{P}\left(\left|\mathrm{FDP}_J^\dagger\left(\zeta_{\alpha-\epsilon}\right) - \mathrm{FDP}_J\left(\zeta_{\alpha-\epsilon}\right)\right| \leq \epsilon, \mathrm{FDP}\left(\zeta_{\alpha-\epsilon}\right) \leq \alpha-\epsilon\right) \\
&= \mathbb{P}\left(\mathrm{FDP}\left(\zeta_{\alpha-\epsilon}\right) \leq \alpha-\epsilon\right) - \mathbb{P}\left(\left|\mathrm{FDP}_J^\dagger\left(\zeta_{\alpha-\epsilon}\right) - \mathrm{FDP}_J\left(\zeta_{\alpha-\epsilon}\right)\right| > \epsilon, \mathrm{FDP}\left(\zeta_{\alpha-\epsilon}\right) \leq \alpha-\epsilon\right) \\
&\geq \mathbb{P}\left(\mathrm{FDP}\left(\zeta_{\alpha-\epsilon}\right) \leq \alpha-\epsilon\right) - \mathbb{P}\left(\left|\mathrm{FDP}_J^\dagger\left(\zeta_{\alpha-\epsilon}\right) - \mathrm{FDP}_J\left(\zeta_{\alpha-\epsilon}\right)\right| > \epsilon\right) \\
&\geq 1 - \epsilon,
\end{aligned}
\tag{A.54}
$$

for $J$ large enough. Thus, we have

$$
\mathbb{P}\left(\zeta^\star \geq \zeta_{\alpha-\epsilon}\right) \geq 1 - \epsilon.
\tag{A.55}
$$

Conditioning on the event $\zeta^\star \geq \zeta_{\alpha-\epsilon}$, we have

$$\limsup_{n,J\to\infty} \mathbb{E}\left[\mathrm{FDP}_J\left(\zeta^\star\right)\right] \leq \limsup_{n,J\to\infty} \mathbb{E}\left[\mathrm{FDP}_J\left(\zeta^\star\right) \mid \zeta^\star \geq \zeta_{\alpha-\epsilon}\right] \mathbb{P}\left(\zeta^\star \geq \zeta_{\alpha-\epsilon}\right) + \epsilon$$

$$\leq \limsup_{n,J\to\infty} \mathbb{E}\left[\left|\mathrm{FDP}_J\left(\zeta^\star\right) - \overline{\mathrm{FDP}}_J\left(\zeta^\star\right)\right| \mid \zeta^\star \geq \zeta_{\alpha-\epsilon}\right] \mathbb{P}\left(\zeta^\star \geq \zeta_{\alpha-\epsilon}\right)$$

$$+ \limsup_{n,J\to\infty} \mathbb{E}\left[\left|\mathrm{FDP}_J^\dagger\left(\zeta^\star\right) - \overline{\mathrm{FDP}}_J\left(\zeta^\star\right)\right| \mid \zeta^\star \geq \zeta_{\alpha-\epsilon}\right] \mathbb{P}\left(\zeta^\star \geq \zeta_{\alpha-\epsilon}\right)$$

$$+ \limsup_{n,J\to\infty} \mathbb{E}\left[\mathrm{FDP}_J^\dagger\left(\zeta^\star\right) \mid \zeta^\star \geq \zeta_{\alpha-\epsilon}\right] \mathbb{P}\left(\zeta^\star \geq \zeta_{\alpha-\epsilon}\right) + \epsilon$$

$$\leq \limsup_{n,J\to\infty} \mathbb{E}\left[\sup_{\zeta\in[\zeta_{\alpha-\epsilon},1]} \left|\mathrm{FDP}_J(\zeta) - \overline{\mathrm{FDP}}_J(\zeta)\right|\right]$$

$$+ \limsup_{n,J\to\infty} \mathbb{E}\left[\sup_{\zeta\in[\zeta_{\alpha-\epsilon},1]} \left|\mathrm{FDP}_J^\dagger(\zeta) - \overline{\mathrm{FDP}}_J(\zeta)\right|\right]$$

$$+ \limsup_{n,J\to\infty} \mathbb{E}\left[\mathrm{FDP}_J^\dagger\left(\zeta^\star\right)\right] + \epsilon.$$

$$\text{(A.56)}$$

The first two terms are 0 based on Lemma 2 and the dominated convergence theorem. For the third term, we have $\mathrm{FDP}_J^\dagger\left(\zeta^\star\right) \leq \alpha$ by the definition of $\zeta^\star$. This concludes the proof of Theorem 2. $\quad\square$

