# OpenReview forum: "AMDP: An Adaptive Detection Procedure for False Discovery Rate Control in High-Dimensional Mediation Analysis"
_NeurIPS.cc/2023/Conference — NeurIPS 2023 spotlight_

### Official Review · Reviewer_qu3t · 2023-07-07

**Soundness:** 3 good
**Presentation:** 3 good
**Contribution:** 3 good
**Rating:** 7
**Confidence:** 3

**Summary:**

This paper proposes a high-dimensional mediation analysis detection procedure with false discovery control. Classic FDR control for multiple testing does not use information across tests making it overly conservative. The authors propose a local FDR-based procedure for identifying mediators and a data-driven approach for estimating the null densities and determining thresholds. Experiments show the proposed approach enjoys high power while still maintaining good FDR control.

**Strengths:**

The application of multiple testing on high-dimensional mediation analysis is interesting.

The local FDR uses across-dimensional information, and the test statistics and thresholds can be estimated from data.

The proposed algorithm maximises power while maintaining FDR control.

Applying a local FDR-based method mediation analysis problem is not trivial. The data-driven approach proposed in Section 2.2 could be useful in other FDR control applications (e.g. how the null densities are obtained).

**Weaknesses:**

This paper is very technical. I would appreciate some more introduction and motivation:

What is local FDR?
Why do we want to use local FDR in mediation analysis?
What may be the obstacles when using local FDR in mediation analysis?

If the authors could present answers to these questions at the beginning, it would make the paper feel more motivated.

**Questions:**

None.

**Limitations:**

Yes

---

> ### Author Rebuttal · Authors · 2023-08-09
>
> Thanks for your valuable suggestions. We will add more introduction and motivation in the revised version. Now we respond the comments raised by you point-by-point.
>
> **Question 1:**
>
> * What is local FDR?
>
> **Response:** In the context of multiple hypothesis testing, the false discovery rate (FDR) is a measure that quantifies the proportion of false positives among the total number of rejected null hypotheses. The local false discovery rate takes this concept one step further and aims to estimate the false discovery rate for each individual test or hypothesis separately (Efron et al., 2001). It represents the posterior probability that a hypothesis is null, given its corresponding p-value. The local FDR plays an important role in multiple hypothesis testing as it considers the varying degrees of uncertainty and statistical evidence supporting or refuting each individual hypothesis, and it allows for a more nuanced and accurate evaluation of the significance of each hypothesis by leveraging information across large-scale tests.
>
> **Question 2:**
>
> * Why do we want to use local FDR in mediation analysis?
>
> **Response:** High-dimensional mediation analysis is often associated with a multiple testing problem for detecting significant mediators. Assessing the uncertainty of this detecting process via false discovery rate (FDR) has garnered great interest. To control the FDR in multiple testing, _**two essential steps are involved: ranking and selection.**_ Existing approaches either construct p-values without calibration or disregard the joint information across tests, leading to conservation in FDR control or non-optimal ranking rules for multiple hypotheses. In contrast, local FDR allows for assessing the significance of each individual mediation path separately while considering the multiple testing aspect. Moreover, it produces the optimal rule for ranking hypotheses, as demonstrated in Theorem 1 of the manuscript. In summary, the proposed local FDR improves the accuracy and reliability of identifying significant mediation paths between variables in multiple hypothesis testing.
>
>
> **Question 3:**
>
> * What may be the obstacles when using local FDR in mediation analysis?
>
> **Response:** The main challenge we encounter pertains to estimate the local FDR within the framework of mediation analysis. In this context, the density of p-values follows a mixture distribution, as indicated in Equation (4). This mixture distribution involves two distinct types of null hypotheses: $H_{01}$ (corresponding to $f_{01}(p)$) and $H_{10}$ (corresponding to $f_{10}(p)$). Effectively distinguishing between these two components and obtaining precise estimations for $f_{10}(p)$ and $f_{01}(p)$ present a formidable obstacle. Motivated by the knockoff method (Barber et al., 2015), we explore a strategy rooted in the symmetry property exhibited by p-values under the composite null hypothesis, see lines 183-194 for more details.
>
> We will rewrite our introduction section and integrate the motivation above in revised version to enrich the readability of the article.
>
> **Reference:**
>
> Barber, R. F., & Candès, E. J. (2015). Controlling the false discovery rate via knockoffs. The Annals of Statistics, 43(5), 2055 – 2085.
>
> Efron, B., Tibshirani, R., Storey, J. D., & Tusher, V. (2001). Empirical Bayes analysis of a microarray experiment. Journal of the American statistical association, 96(456), 1151-1160.

---

> > ### Comment · Reviewer_qu3t · 2023-08-14
> >
> > I have read the authors' replies and encourage authors to revise their papers as they mentioned in the reply.

---

> > > ### Author Response · Authors · 2023-08-14
> > > **Response to reviewer**
> > >
> > > Thank you very much for your careful review and constructive comments, which have much contributed to improving the paper.

---

### Official Review · Reviewer_ZQdD · 2023-07-11

**Soundness:** 3 good
**Presentation:** 2 fair
**Contribution:** 3 good
**Rating:** 6
**Confidence:** 2

**Summary:**

This paper describes a method to control for false discoveries in high-dimensional mediation analysis. This is the problem of inferring whether any of a large set of possible mediator variables act as a mediator between an exposure variable and an outcome variable. This work is particularly interested in the application where the exposure is a single-nucleotide polymorphism (SNP), the exposure is the expression level of a gene, and the mediators are the methylation levels at a set of CpGs. It is known that methylation is involved in the regulation of gene expression, so in this setting one wishes to discover whether the SNP is affecting the methylation at a CpG and in turn this is affecting the level of gene expression of a gene. This is an interesting problem and one of interest when seeking to uncover the functional effect of genetic variants. In this setting, there will typically be many CpGs as possible mediators, so we are in the realm of high-dimensional statistics and need to account for the fact that we are running a large number of tests in parallel. Previous methods for this problem are deemed to be too conversatve, sacrificing power for the suppression of false discoveries, and the authors seek to identify an approach which has improved power.

The method is verified on simulated data and illustrated on a prostate cancer data-set, where it is shown to identify more (SNP, CpG, gene) triples than existing methods

**Strengths:**

- The paper tackles a problem of practical interest
- The method is justified theoretically (I did not check the proofs)
- The method is justified empirically on simulated data and illustrated on a real-world data set

**Weaknesses:**

The presentation felt a bit compressed with many relevant details deferred to the appendix, such as the examples described in Remark 2 and many of the plots relating to the prostate cancer data-set.

The authors provide limited information on the context, assuming a reader will be familiar with genetic and epigenetic analysis.

**Questions:**

No questions

**Limitations:**

I do not believe that the method has any obvious negative societal impacts.

---

> ### Author Rebuttal · Authors · 2023-08-09
>
> **Question 1:**
>
> * The presentation felt a bit compressed with many relevant details deferred to the appendix, such as the examples described in Remark 2 and many of the plots relating to the prostate cancer dataset.
>
> **Response:** Thanks for you carefully to review our paper. We will make adjustments to the typesetting of both the main text and appendix, as well as revise and go through the content in the updated version.
>
> **Question 2:**
>
> * The authors provide limited information on the context, assuming a reader will be familiar with genetic and epigenetic analysis.
>
> **Response:** Thanks for your insightful comments to improve the quality and depth of our work. In the revised version, we will expand on the information in the field of genetics and epigenetics, offering a more detailed and comprehensive explanation.

---

### Official Review · Reviewer_9J5U · 2023-07-13

**Soundness:** 3 good
**Presentation:** 2 fair
**Contribution:** 3 good
**Rating:** 6
**Confidence:** 1

**Summary:**

The paper proposes and analyses a new method for FDR control.

**Strengths:**

The paper is clear on the method, the assumptions and the proofs. The empirical section is promising and adequate.


**Weaknesses:**

Some of the assumptions and the writing are not clear, or seem too strong to me. For example the assumption about "no confounding" which is not listed as a formal assumption but is mentioned in the text, and attributed to another paper. It would be helpful is the authors could provide an explanation about why this is not a limiting assumption.

The assumption 2 seems reasonable, but assumption 1 again seems limiting, with a few distributions that could satisfy it. It is not a paper breaking assumption, but it would be useful if the authors are upfront about it, and discuss which possible distributions satisfy it. Similarly, about assumption 2 it would be helpful if the authors could write a couple of sentences discussing what the assumption means and implies.

The abstract and the intro makes a big deal about the curse of dimensionality becoming a blessing, but this was not obvious to me from the paper's text. could the authors explain what they mean by this exactly ? could you point out where this is discussed in the paper?

I thought the "limitations" section was required this year for neurips submissions. but this is missing, and the discussions section at the end is not comprehensive about this.

**Questions:**

Please see the weakness section.

**Limitations:**

No.

---

> ### Author Rebuttal · Authors · 2023-08-09
>
> We thank reviewer 9J5U for your comments and for citing your concerns. 9J5U’s main concern is that some assumptions in our paper seem too strong. We can provide some explanations.
>
> Major questions:
>
> **Question 1:**
>
> * The reviewer concern about the assumption about "no confounding".
>
> **Response:** To be more precise, this assumption is well known as the sequential ignorability assumption, as established in the modern causal inference framework (Imai et al., 2010; Valeri et al., 2013). Specifically, this assumption comprises two components: (1) "no unmeasured confounding" of the exposure-outcome relationship and (2) "no unmeasured confounding" of the mediator-outcome relationship. Under the sequential ignorability assumption, the Natural Indirect Effect (NIE) can be identified and is equivalent to the product of $\alpha$ and $\beta$ in our model (Imai et al., 2010). In the revised version, we have taken the reviewer's feedback into account and further clarified the language by switching "no confounding" to "no unmeasured confounding" in Section 2.1. This change helps ensure a clearer and more precise description of our assumptions and their implications for the mediation analysis.
>
> Although the sequential ignorability assumption is a common assumption in traditional mediation analysis, it's worth noting that certain studies have delved into the intricacies of the hidden confounder issue, as mentioned in Song et al. (2020) and Song et al. (2021). Given this perspective, we will further consider extending our current work to settings where hidden confounders are present.
>
> **Question 2:**
>
> * The reviewer asked for explanations of Assumptions 1-2.
>
> **Response:** Our method extracts a pair $(p_1, p_2)$ for each exposure-mediator-outcome relationship and employs these pairs to estimate the FDP on a two-dimensional plane $[0,1] \times [0,1]$. The theoretical basis supporting FDP estimation is the assumption that p-values are uniformly distributed under the null hypothesis, a widely recognized principle (Benjamini et al., 1995; Hung et al., 1997). Due to the presence of a composite null hypothesis in the mediation effect, we elaborate on Assumptions 1-2 to illustrate the properties of the p-value distribution under composite null hypothesis.
>
> (1) For Assumption 1, under $H_{00}$, both $p_1$ and $p_2$ obey the uniform distribution, resulting in $(p_1, p_2)$ also following the uniform distribution on the two-dimensional plane $[0,1]\times[0,1]$. Consequently, the sampling distribution of $(p_1, p_2)$ is symmetrical around $p_1=0.5$ and $p_2=0.5$. Under $H_{01}$, $p_1$ still obeys the uniform distribution, but $p_2$ does not, leading to $(p_1, p_2)$ being only symmetrical about $p_1=0.5$ on $[0,1]\times[0,1]$. Similarly, under $H_{10}$, $p_2$ obeys the uniform distribution, resulting in $(p_1, p_2)$ being only symmetrical about $p_2=0.5$ on $[0,1]\times[0,1]$. It is essential to emphasize that Assumption 1 specifically applies to the null mediators.
>
> (2) For Assumption 2, since a non-null p-value theoretically lies within $[0, 0.5)$, it implies that the rejection region $S$ is a subset of $[0, 0.5) \times[0, 0.5)$ without requiring additional information. Consequently, $S$ and its symmetric regions do not overlap. Moreover, as the sample size $n$ tends to infinity, the probability of p-values under alternative hypotheses falling within $[0.5, 1]$ approaches zero. For example, as $n$ goes to infinity, p-values under $H_{11}$ and $H_{10}$ are not expected to fall within the region $\tilde{S_{01}}$ because the region $\tilde{S_{01}}$
> theoretically only contains p-values under $H_{00}$ and $H_{01}$. Similarly, the region $\tilde{S_{10}}$
> theoretically only includes p-values under $H_{00}$ and $H_{10}$.
>
>
> **Question 3:**
>
> * The reviewer asked for explanation of “the curse of dimensionality becoming a blessing”.
>
> **Response:** The core idea behind our method is to extract a set of $(p_1, p_2)$ for each exposure-mediator-outcome relationship and constructing a p-value-based statistic, which we refer to as the local FDR. We can estimate the local FDR by treating each set of p-values as a sample point. It effectively converts the high-dimensional problem into a large sample scenario. The increasing dimensionality yields more accurate estimates of the local FDR involving the nonparametric density estimation. Consequently, as the dimension increases, we can better control the FDR, making our method more effective in practical applications.
>
> **Question 4:**
> * The reviewer asked for "limitations" section.
>
> **Response:** As previously discussed in Question 3, our article focuses on addressing high-dimensional mediation effects. Consequently, the estimation of local FDR and FDP might not be as effective for low-dimensional mediation problems. We will rewrite our discussion section and incorporate the limitation in revised version to provide a more comprehensive analysis.
>
> **Reference:**
>
> Benjamini, Y., & Hochberg, Y. (1995), ..., Journal of the Royal statistical society: series B (Methodological), 57(1), 289-300.
>
> Hung, H. J., O'Neill, R. T., Bauer, P., & Kohne, K. (1997), ..., Biometrics, 11-22.
>
> Imai, K., Keele, L., & Tingley, D. (2010), ..., Psychological methods, 15(4), 309.
>
> Song, Y., Zhou, X., Zhang, M., Zhao, W., Liu, Y., Kardia, S. L., ... & Mukherjee, B. (2020), ..., Biometrics, 76(3), 700-710.
>
> Song, Y., Zhou, X., Kang, J., Aung, M. T., Zhang, M., Zhao, W., ... & Mukherjee, B. (2021), ..., Statistics in medicine, 40(27), 6038-6056.
>
> Valeri, L., & VanderWeele, T. J. (2013), ..., Psychological methods, 18(2), 137.

---

> > ### Comment · Reviewer_9J5U · 2023-08-16
> >
> > THank you for addressing my comments, I will update my score. I will keep the low confidence, since this is not my area and I am not very confident about the specific contributions.

---

> > > ### Author Response · Authors · 2023-08-17
> > > **Response to Reviewer 9J5U**
> > >
> > > Thank you very much for the response!

---

### Official Review · Reviewer_oiJi · 2023-07-14

**Soundness:** 3 good
**Presentation:** 2 fair
**Contribution:** 3 good
**Rating:** 6
**Confidence:** 3

**Summary:**

UPDATE

In light of the revisions the authors made I have raised my score and now support acceptance of this work. Thank you very much.

-----------------------------------------
The authors propose a procedure to increase statistical power to identify mediators while controlling the FDR in high-dimensional data sets. Their method leverages the proportions of composite null hypotheses and the distribution of p-values under the alternative to derive an algorithm to estimate p-values for mediator variables while controlling the FDR. They perform a theoretical analysis suggesting that their method is asymptotically optimal and showcase that it controls FDR more consistently than other methods (DACT, JC, JS-mixture) in a simulation study and identifies more mediators than JS-mixture in an empirical study suggesting higher statistical power.

**Strengths:**

- The problem is very relevant as high-dimensional data are frequent not only in genetics, but also in other domains like neuroscience and user data, which all potentially suffer from similar problems.
- The work is very extensive comprising theoretical analysis, simulations and an empirical application of the methods.
- Related methods are extensively discussed in the introduction.


**Weaknesses:**

- There is no real discussion of the limitations/breakpoints of this method.
- The effect sizes chosen for the simulation analysis seem to be unreasonably large (at least showing how the method behaves for smaller effect sizes, which are more commonly observed empirically, would be informative).
- The paper is occasionally quite dense and would benefit from focusing on some key aspects.


**Questions:**

This is very interesting work with broad applications to many empirical studies. I am willing to support acceptance of this paper, if the following concerns/questions are addressed appropriately. I hope you find my comments helpful and constructive.

**Major points**
- Can you provide a motivation for the choice of simulation parameters (i.e., sample sizes, number of mediators, FDR alpha and effect sizes). Current genetic studies tend to have larger sample sizes ( in the order of 20.000) while older studies had smaller sample sizes so including a wider range of samples e.g. 100, 500, 1000, 10000, 20000 may better cover empirical scenarios. Moreover, I don’t understand why you picked alpha = 0.1 as reference, since most studies choose alpha=0.05, for genetic studies this might even be smaller.
- In Figure 1, it appears that AMDP deviates more from the 0.1 FDR line when the effect sizes get smaller. Could you speculate on why that is? On that note, the effect sizes chosen are rather large. What happens for smaller effect sizes (up to .2 or .3). Does this deviation get larger, is this a breakpoint for the method?
- Similarly, in the empirical analysis (Figure 3), JS-mixtures identifies more triplets for alpha=0.01. Could you explain why that is, does this trend hold for alphas that are even smaller like 0.001?
- What are the conditions under which the method is not appropriate? When does it break? I am missing a clear limitation section.


**Minor point**
- p.8 ll. 288 ff. essentially contains a figure caption of a figure that is presented in the supplement, which is not very helpful for the reader. I would suggest to either summarize the take-aways from this supplementary analysis here and move the caption to the figure where it belongs or spend this space on discussion the implications including break points and limitations of the analyses you present in the main manuscript.
- Just out of curiosity, can you elaborate on how this method could be used for image analysis? Can this be applied to neuroimaging (e.g., fMRI) as well?



**Limitations:**

- The paper does not have a clear limitation section (see Questions).

---

> ### Author Rebuttal · Authors · 2023-08-09
>
> Thanks for the thoughtful comments.
>
> **Weaknesses 1：** Thanks for your reminder. Our method is designed to high-dimensional mediation analysis, but its performance may fail in the low-dimensional designs. We will rewrite our discussion section and incorporate the limitation in revised version to provide a more comprehensive analysis.
>
> **Weaknesses 2：** We adopt the similar settings as these in Dai et al. (2023).  More details about the parameter settings can be found in our reply to Question 1.
>
> **Weaknesses 3：** Your suggestion will lead to a significant improvement of an early manuscript. The high-dimensional mediation analysis is often associated with a multiple testing problem for detecting significant mediators, which has attracted significant interest. _**To control the FDR, our article will focus on two key steps: ranking and selection.**_  We will emphasize the two key aspects in revised version to enhance the readability of the article.
>
>
> **Question 1:**
>
> * The reviewer's inquiry regarding the motivation for the choice of simulation parameter.
>
> **Response:**
>
> (1) In Figures 1-2, we assess how the four methods (JS-mixture, DACT (Efron), DACT (JC), and AMDP) are influenced by effect size, the large mediator size and sample size.
>
> (2) We aimed to closely simulate real-world data scenarios. (a) We referred to several real datasets including the TCGA lung cancer cohort dataset (Zhang et al., 2021), the Multi-Ethnic Study of Atherosclerosis (Du et al., 2023), and the TCGA prostate cancer dataset (Dai et al., 2023), and then construct the simulation examples. (b) We adopt the similar parameter settings as these in Dai et al. (2023).
>
> (3) We follow widely used FDR level 0.05 (Song et al., 2020; Guo et al., 2023) and FDR level 0.1 (Dai et al., 2023; Mosig et al., 2021).
>
> * The reviewer asked if we could cover empirical scenarios with a wider range of sample size, and the reason why we pick alpha=0.1 as reference.
>
> **Response:** As the response to Question 1, we will adopt the commonly used FDR levels 0.05 and 0.1. Our method is specifically designed for genetic data, as suggested by Mosig et al. (2001), we adopted FDR level of 0.1 in the original manuscript. Following your suggestion, we have integrated experiments at the FDR threshold of 0.05 with a wide range of sample sizes (200, 500, 1000, 5000).  We present the experimental results under sparse alternatives scenario and dense alternatives scenario in PDF format. From Tables 1-2 in PDF format, we can observe the similarly results as these with the FDR level of 0.1. Note that when the sample size reaches 5000, the power of all four methods converges close to 1, so we do not undertake experiments with larger sample sizes of 10000 and 20000.
>
>
> **Question 2:**
>
> * The reviewer asked the reason why AMDP deviates more from the 0.1 FDR line when the effect sizes get smaller in Figure 1.
>
> **Response:** There is the missing "+1" in Equation (10). To enhance the robustness of FDR control, our method estimates the numerator of Equation (10) as (the number of false discoveries + 1), which was used in Du et al. (2023) and Guo et al. (2022). When the effect size is smaller, it yields a small denominator for the FDP estimation. Thus, the "+1" term becomes non-negligible, leading to slightly conservative FDR controls. To mitigate this problem, the "+0.5" adjustment can be employed effectively.  As depicted in Figure 1, the JS-mixture method also exhibits a similar phenomenon. The difficulty to detect significant variables with minor effect sizes results in rare cases of no rejection in some experiments, which subsequently reduces the average FDR.
>
> * The reviewer shows concern about the effect sizes chosen in our simulation study.
>
> **Response:** Our simulation settings encompass a wide range of effect sizes, including 0.2, 0.3, and even smaller values such as 0.12. More details can be found in the response to Question 1.
>
> **Question 3:**
>
> * The reviewer’ question about the trend observed in Figure 3 for different alpha levels, including alphas smaller than 0.01.
>
> **Response:** We agree that JS-mixture identifies more triplets than our method when alpha is smaller. As the response in Question 2, our method employs a +1 adjustment in Equation (10) to enhance FDR control. As the alpha level decreases, the number of rejections is significantly reduced, and the +1 term becomes non-negligible, leading to fewer identified triplets. It's essential to point out that people usually care about the results of widely used FDR levels, such as 0.05 and 0.1.
>
> **Question 4:**
>
> * The reviewer asked a clear limitation section.
>
> **Response:** Although the performance of AMDP suffers in low-dimensional designs, as stated in Weaknesses 1, our method does not exhibit a well-defined breakpoint.
>
> **Minor question:**
>
> * **Response:** Thanks for your valuable suggestion. We will incorporate the necessary modifications in the revised version.
>
> * **Response:** Thanks for the interesting question. The application of our method to neuroimaging is a plausible avenue. In voxel-based analyses, our method may be used to capture active voxels in various brain regions, including the visual, auditory, and motor regions. For further inspiration, see Chang et al. (2023).
>
> **Reference:**
>
> Chang, J., He, J., Kang, J., & Wu, M. (2023), …, JASA, 1-14.
>
> Dai, J. Y., Stanford, J. L., & LeBlanc, M. (2022), …, JASA, 117, 198-213.
>
> Du, J., Zhou, X., …, & Mukherjee, B. (2023), …, Genet Epidemiol, 47, 167-184.
>
> Du, L., Guo, X., Sun, W., & Zou, C. (2023), …, JASA, 118, 607-621.
>
> Efron, B. (2004), …, JASA, 99, 96-104.
>
> Guo, X., Ren, H., Zou, C., & Li, R. (2022), …, JASA, 1-13.
>
> Guo, X., Li, R., Liu, J., & Zeng, M. (2023), …, JBES, 1-14.
>
> Mosig, M. O., Lipkin, E., ..., & Friedmann, A. (2001), …, Genetics, 157, 1683-1698.
>
> Song, Y., Zhou, X., ..., & Mukherjee, B. (2020), …, Biometrics, 76, 700-710.
>
> Zhang, H., Zheng, Y., Hou, L., Zheng, C., & Liu, L. (2021), …, Bioinformatics, 37, 3815-3821.

---

> > ### Comment · Reviewer_oiJi · 2023-08-14
> > **Limitation section**
> >
> > Thank you very much for addressing most of my concerns. However, do I understand you correctly, that you do not plan to include a limitation section? If that is the case, I would urge the authors to strongly consider including a limitation section. They best know what these limitations could be, but I seriously doubt that there are no limitations whatsoever. Should this be included, I am willing to raise my score.

---

> > > ### Author Response · Authors · 2023-08-14
> > > **Response to reviewer**
> > >
> > > We thank the reviewer oiJi for the feedback. We genuinely value the reviewer's insights, and are willing to add a limitation section in the revised version as:
> > >
> > > _“Our approach effectively handles high-dimensional mediators but may not perform optimally when confronted with low-dimensional mediators. This distinction is attributed to the nature of our method, wherein the two-dimensional p-values linked to each exposure-mediator-outcome relationship effectively serve as 'samples' for the estimation of local FDR and FDP. Consequently, the reduction in dimensionality can lead to less precise estimates of local FDR and FDP.”_

---

> > > > ### Comment · Reviewer_oiJi · 2023-08-15
> > > > **All concerns addressed and raising score**
> > > >
> > > > Thank you very much. I will raise my score and wish the authors a great conference. Thank you for this interesting work.

---

> > > > > ### Author Response · Authors · 2023-08-15
> > > > > **Thank you!**
> > > > >
> > > > > Thank you again for your response.

---

### Author Rebuttal · Authors · 2023-08-09

Thanks again for the comments. We present additional experimental results under sparse alternatives scenario and dense alternatives scenario with the FDR level of 0.05 in the PDF format.

---

### Comment · Reviewer_ZQdD · 2023-08-14
**Review summary**

I've read the other reviews and author response and would support acceptance should the concerns raised in the reviewers be addressed

---

> ### Author Response · Authors · 2023-08-15
> **Response to reviewer**
>
> We greatly thank the reviewer for the careful reading and useful comments. We are committed to diligently addressing all the reviewers’ concerns and suggestions raised, and we will make sure to incorporate more details and update the final version.

---

### Decision · Program_Chairs · 2023-09-21

**Decision:**

Accept (spotlight)

**Comment:**

The paper discusses an important problem dealing with high-dimensional data in many domains.  Overall, we found the method to be theoretically sound and empirically validated; we think the paper is worthy of being accepted.  However there are several aspects that can be improved.  Please reflect reviewers' comments and the discussions in making a final version.  More specifically, it is desired to include discussions on the parameter choice and assumptions.  Also, be sure to include the limitation as you comment in the discussion.